

# Simultaneous Detection of Ozone and Nitrogen Dioxide by Oxygen Anion Chemical Ionization Mass Spectrometry: A Fast Time Response Sensor Suitable for Eddy Covariance Measurements

Gordon A. Novak, Michael P. Vermeuel, Timothy H. Bertram

Department of Chemistry, University of Wisconsin – Madison, Madison, WI, USA

*Correspondence to*: Timothy H. Bertram (timothy.bertram@wisc.edu)

**Abstract.** We report on the development, characterization, and field deployment of a fast time response sensor for measuring ozone ($O_3$) and nitrogen dioxide ($NO_2$) concentrations utilizing chemical ionization time-of-flight mass spectrometry (CI-ToFMS) with oxygen anion ($O_2^-$) reagent ion chemistry. We demonstrate that the oxygen anion chemical ionization mass
spectrometer (Ox-CIMS) is highly sensitive to both $O_3$ (180 ions s$^{-1}$ pptv$^{-1}$) and $NO_2$ (97 ions s$^{-1}$ pptv$^{-1}$),  corresponding to detection limits (3σ, 1 s averages) of 13 and 9.9 pptv, respectively. In both cases, the detection threshold is limited by the magnitude and variability in the background determination. The short-term precision (1 s averages) is better than 0.3% at 10 ppbv $O_3$ and 4% at 10 pptv $NO_2$. We demonstrate that the sensitivity of the $O_3$ measurement to fluctuations in ambient water vapor and carbon dioxide is negligible for typical conditions encountered in the troposphere. The application of the Ox-CIMS
to the measurement of $O_3$ vertical fluxes over the coastal ocean, via eddy covariance (EC), was tested during summer 2018 at Scripps Pier, La Jolla CA. The observed mean ozone deposition velocity ($v_d(O_3)$) was 0.011 cm s$^{-1}$ with a campaign ensemble limit of detection (LOD) of 0.0042 cm s$^{-1}$ at the 95% confidence level, from each 27-minute sampling period LOD. The campaign mean and one standard deviation range of $O_3$ mixing ratios were 38.9 ± 12.3 ppbv. Several fast ozone titration events from local NO emissions were sampled where unit conversion of $O_3$ to $NO_2$ was observed, highlighting instrument utility as a
total odd oxygen ($O_x = O_3 + NO_2$) sensor. The demonstrated precision, sensitivity, and time resolution of this instrument highlight its potential for direct measurements of $O_3$ ocean–atmosphere and biosphere–atmosphere exchange from both stationary and mobile sampling platforms.

## 1 Introduction

The deposition of $O_3$ to the ocean surface is a significant component of the tropospheric ozone budget. Global chemical
transport model studies that explicitly treat $O_3$ deposition, indicate that approximately one-third of total ozone dry deposition is to water surfaces (Ganzeveld et al., 2009)  However, the magnitude of total annual global ozone deposition to ocean surfaces is highly sensitive to the deposition velocity parameterization used, with model estimates ranging from 95 to 360 Tg yr$^{-1}$ (Ganzeveld et al., 2009; Luhar et al., 2017). Several common global chemical transport models including GEOS-Chem (Bey et al., 2001), MOZART-4 (Emmons et al., 2010), and CAM-chem (Lamarque et al., 2012) apply a globally uniform deposition
velocity ($v_d$) that ranges between 0.01–0.05 cm s$^{-1}$ depending on the model. In comparison to terrestrial measurements, where



O$_3$ dry deposition velocities are relatively fast (>0.1 cm s$^{-1}$, (Zhang et al., 2003)), there is a paucity of direct observations of ozone deposition to the ocean surface necessary to constrain atmospheric models. Previous studies of O$_3$ deposition to water surfaces have been made from coastal towers (Gallagher et al., 2001), aircraft (Faloona et al., 2005; Kawa and Pearson, 1989; Lenschow et al., 1981), underway research vessels (Helmig et al., 2012), and in the laboratory (McKay et al., 1992), with

observed $v_d$(O$_3$) ranging between 0.01 and 0.15 cm s$^{-1}$. To date there is no consensus on whether measured O$_3$ deposition velocities show a wind speed dependence (Fairall et al., 2007). The most comprehensive dataset is from Helmig et al. (2012), which reported a deposition velocity range of 0.009 – 0.034 cm s$^{-1}$ from 1700 hours of observation over five research cruises. This dataset showed variability of $v_d$(O$_3$) with wind speed ($U_{10}$) and sea–surface temperature (SST), highlighting the need for further field observations as constraints for model parameterizations.

40         The small magnitude of O$_3$ ocean–atmosphere vertical fluxes presents a significant analytical challenge for existing ozone sensors used in eddy covariance (EC) analyses. Driven in part by stringent sensor requirements for EC techniques, significant uncertainties in the magnitude and variability of ozone deposition to water surfaces remain. In contrast, O$_3$ vertical fluxes to terrestrial surfaces are 10 to 100 times faster than to water surfaces, significantly loosening sensor precision requirements. Nonetheless, significant variability in $v_d$(O$_3$) exists between surface types (e.g. soil *vs.* leaf) (Wesely and Hicks,

2000). Terrestrial deposition velocities also show strong diel and seasonal variability due to factors such as stomatal opening and within-canopy chemistry (Fares et al., 2010; Fowler et al., 2001; Kurpius and Goldstein, 2003). Highly accurate and precise measurements of O$_3$ are required to correctly model the response of $v_d$(O$_3$) to each of these factors. While terrestrial and ocean exchange studies have substantial differences in experimental design, a sensor suitable for ocean–atmosphere ozone deposition measurements *via* EC is expected to be highly capable of biosphere–atmosphere measurements due to the significantly larger

deposition rates and similar accuracy requirements.

         Eddy covariance measurements typically require fast (1-10 Hz), high precision sensors in order to resolve covariance on the timescales of the fastest atmospheric turbulent eddies. Due to this constraint, ozone flux measurements have primarily utilized fast response chemiluminescence sensors. Chemiluminescence detectors can use either wet, dry, or gas-phase reagents for detection with important differences between them (Muller et al., 2010). Gas-phase chemiluminescence sensors are

typically based on the reaction of O$_3$ with nitric oxide (NO) to form an excited state NO$_2^*$ which then relaxes to the ground state, emitting a photon that can be detected. This method has well understood reaction kinetics and allows for high sensitivity detection on the order of 2.8 counts s$^{-1}$ pptv$^{-1}$ (Bariteau et al., 2010; Pearson, 1990). A practical disadvantage to this technique is the necessity of a compressed cylinder of NO which is highly toxic. Wet chemiluminescence techniques are used less, as they exhibit generally lower sensitivity than dry chemiluminescence sensors and can be limited by issues in the liquid flow

(Keronen et al., 2003).

         Dry chemiluminescence sensors have the simplest operation and have seen the most regular use for EC studies (Güsten et al., 1992; Tuovinen et al., 2004). However, dry chemiluminescence sensor discs require conditioning with high ozone (up to 400 ppbv for several hours) before operation, are known to degrade over time, and have high variability in sensitivity between sensor discs (Weinheimer, 2007). These factors have led to limitations in long term stability and to





uncertainty in calibration factors for dry chemiluminescence sensors, resulting in uncertainty in the accuracy of the flux measurement (Muller et al., 2010). Muller et al.(2010), also reported a comparison of two identical co-located dry chemiluminescence sensors with half-hourly flux values differing by up to a factor of two and a mean hourly flux difference ranging from 0 to 23% between sensors. Recently Zahn et al., (2012) reported the development of a commercial dry chemiluminescence ozone detector capable of fast (>10 Hz) measurements with high sensitivity (~9 counts s$^{-1}$ pptv$^{-1}$) suitable

for EC or mobile platform sampling. However, they also report issues of short- and long-term drift and variability between sensor discs. These accuracy and drift concerns have driven an interest in the development of a new, stable and fast ozone sensor suitable for EC measurements from both stationary and mobile sampling platforms.

      In addition to the inherently small magnitude of $v_d(O_3)$, the fast chemical titration of $O_3$ by NO (R1) often complicates the interpretation of $v_d(O_3)$ measurements. Surface emissions of NO result in a high bias in the measured deposition velocity

when the titration reaction (R1) is fast relative to the transport time to the height of the sensor.

$$O_3 + NO \rightarrow NO_2 + O_2 \qquad k(O_3+NO) = 1.8 \times 10^{-14} \text{ cm}^3 \text{ molecule}^{-1} \text{ s}^{-1} \qquad \text{(R1)}$$

Surface NO emissions from both biogenic and anthropogenic sources are widespread, with ocean emissions on the order of 1 $\times$ 10$^8$ molecules cm$^{-2}$ s$^{-1}$ (Zafiriou and McFarland, 1981) and soil emissions ranging from 5 $\times$ 10$^9$ to 2 $\times$ 10$^{11}$ molecules cm$^{-2}$ s$^{-1}$ (Yienger and Levy, 2004). These emissions correspond to a positive bias in the observed $v_d(O_3)$ dry deposition rate on the

order of 5% in the marine atmosphere (discussed in section 3.7.1) and up to 50% in a forested site (Dorsey et al., 2004). Simultaneous flux detection of $O_3$ with one or both of NO or $NO_2$ is commonly used to address this flux divergence problem (Finco et al., 2018; Stella et al., 2013). However, these studies typically require separate sensors for $O_3$ and $NO_x$ which can introduce additional sources of uncertainty. Related challenges of fast $O_3$ titration exists for quantification of $O_3$ from mobile platforms where there is dynamic sampling of different airmasses with potentially differing $O_3$–NO–$NO_2$ steady-state

conditions.

      In what follows, we describe the characterization and first field observations of a novel oxygen anion chemical ionization time-of-flight mass spectrometer (Ox-CIMS) sensor for $O_3$ and $NO_2$. Over the past two decades, chemical ionization mass spectrometry (CIMS) techniques have emerged as sensitive, selective, and accurate detection methods for a diverse suite of reactive trace gases (Huey, 2007). Successful application of CIMS for EC flux measurements have been demonstrated from

many sampling platforms including ground sites (Kim et al., 2014; Nguyen et al., 2015), aircraft (Wolfe et al., 2015), and underway research vessels (Blomquist et al., 2010; Kim et al., 2017; Yang et al., 2013) employing a variety of reagent ion chemistry systems. Here we demonstrate the suitability of the Ox-CIMS for EC flux measurements and provide detailed laboratory characterization of the instrument.



## 2 Laboratory Characterization

### 2.1 Chemical-ionization time-of-flight mass spectrometer

A complete description of the CI-ToFMS instrument (Aerodyne Research Inc., TOFWERK AG) can be found in Bertram et al. (2011). In what follows we highlight significant differences in the operation of the instrument from what is discussed in Bertram et al., (2011). Oxygen anions are generated by passing an 11:1 volumetric blend of Ultrahigh Purity (UHP) $N_2$ and $O_2$ gas (both Airgas 5.0 grade) through a polonium-210 α-particle source (NRD, P-2021 Ionizer). This $N_2:O_2$ volume ratio was found empirically to maximize total reagent ion signal in our instrument while minimizing background signal at the $O_3$ detection product ($CO_3^-$,– 60 m/$Q$). Further discussion of the reagent ion chemistry and precursor concentration can be found in sections 2.2 and 2.8. The reagent ion stream then mixes with ambient air in an ion-molecule reaction (IMR) chamber held at 95 mbar where product ions were generated. Further discussion of the dependence of instrument sensitivity on IMR pressure can be found in section 2.6. At this pressure, the residence time in the IMR is estimated to be on the order of 100 ms. Product ions then pass into three differentially pumped chambers before reaching the ToF mass analyzer. Ions first move from the IMR to a collisional dissociation chamber (CDC) held at 2 mbar which houses a short-segmented RF-only quadrupole ion guide. Field strengths in the IMR and CDC were tuned to be as soft as possible to preserve the transmission of weakly bound clusters while still maintaining acceptable total ion signals (ion optic potentials are listed in Table S1). Ions then sequentially pass into a second RF-only quadrupole chamber held at $1.4 \times 10^{-2}$ mbar and a final chamber containing focusing optics which prepare the ion beam for entry into the ToF mass analyzer (TOFWERK AG and Aerodyne Research Inc.). The mass resolving power ($M/\Delta M$) of the instrument as configured for these experiments was greater than 900 at –60 m/$Q$. All ion count rates reported here are for unit mass resolution integrated peak areas. In this work extraction frequencies of 75 kHz were used, resulting in mass spectra from 27-327 –m/$Q$. All mass spectra were saved at 10 Hz for analysis.

### 2.2 Oxygen Anion Chemistry

Oxygen anion ($O_2^-$) reagent ion chemistry has been investigated previously for its use in the detection of nitric acid and more recently hydrogen peroxide (Huey, 1996; O'Sullivan et al., 2018; Vermeuel et al., 2019). Oxygen anion chemistry has also been used for chemical analysis of aerosol particles in a thermal desorption instrument, primarily for detection of particle sulfate and nitrate (Voisin et al., 2003). Oxygen anion chemistry has also been used for the detection of $SO_2$ *via* a multi-step ionization process where $CO_3^-$ reagent ions are generated by the reaction of $O_2^-$ with excess $O_3$ in the presence of $CO_2$. The $CO_3^-$ product then ligand switches with $SO_2$ to form $SO_3^-$ which then quickly reacts with ambient $O_2$ to form the primary detected $SO_5^-$ product (Porter et al., 2018; Thornton et al., 2002a). Ionization of analytes by oxygen anion reagent ion chemistry proceeds through both charge transfer (R2) and adduct formation (R3).

$$O_2(H_2O)_n^- + A \rightarrow O_2(H_2O)_n + A^- \tag{R2}$$
$$O_2(H_2O)_n^- + B \rightarrow B \cdot O_2(H_2O)_n^- \tag{R3}$$


It is expected that charge transfer from oxygen will occur to any analyte with an electron affinity (E.A.) greater than $O_2$ (0.45 eV, (Ervin et al., 2003)) resulting in a relatively non-specific reagent ion chemistry which is sensitive to a wide class of molecules. Adduct formation is observed when the binding enthalpy of the adduct is larger than that of the oxygen-water adduct and the adduct is stable enough to be preserved through the ion optics. This adduct formation framework is analogous to what has been shown for iodide reagent ion chemistry (Lee et al., 2014).

The $O_2^-$ reagent ions present in the IMR are expected to have a series of attached water molecules at ambient humidity and the IMR pressure (95 mbar) and electric field strengths used in this study (Bork et al., 2011). The reagent ion is therefore reported as $O_2(H_2O)_n^-$ for the remainder of this work. In the recorded mass spectra from our instrument, all reagent ion signal is observed as $n = 0–1$ (i.e., $O_2^-$ and $O_2(H_2O)^-$) as seen in Fig. 1. Oxygen anion-water clusters larger than $n = 1$ are likely present in the IMR but $H_2O$ evaporates off of the cluster in the CDC before detection due to the lower binding enthalpy of each additional

water in $O_2(H_2O)_n^-$ (Bork et al., 2011) and the high filed strength at the exit of the CDC (Brophy and Farmer, 2016). Variability in the number of attached water molecules ($n$) as a function of humidity introduces the possibility of a water dependence on the ion chemistry, which is discussed further in Section 2.5.

    The detection of ozone ($O_3$) by oxygen anion reagent ion chemistry proceeds *via* a two-step reaction leading to the formation of a carbonate anion ($CO_3^-$), which is the final detected product. First, the oxygen anion ($O_2(H_2O)_n^-$) either transfers

an electron to ozone forming $O_3^-$ (R4a) or forms a stable cluster with ozone (R4b). The ozone anion (either bare or as a cluster with $O_2(H_2O)_n$) then reacts with a neutral $CO_2$ molecule to form $CO_3^-$ (R5a–5b) which is the primary, detected product in the mass spectrometer. The electron affinity of $O_3$ is 2.1 eV (Arnold et al., 1994).

$$O_3 + O_2(H_2O)_n^- \rightarrow O_3^- + O_2 + nH_2O \tag{R4a}$$

$$O_3 + O_2(H_2O)_n^- \rightarrow O_2(O_3)(H_2O)_n^- \tag{R4b}$$

$$O_3^- + CO_2 \rightarrow CO_3^- + O_2 \tag{R5a}$$

$$O_2(O_3)(H_2O)_n^- + CO_2 \rightarrow CO_3^- + 2O_2 + nH_2O \tag{R5b}$$

    It is not clear whether it is the bare ozone anion (R4a & R5a) or the cluster (R4b & R5b) that goes on to react with $CO_2$ to form the carbonate anion. The $O_2(O_3)(H_2O)_n^-$ product has not been observed in the mass spectrometer, but it may exist in the IMR and dissociate as it transfers into the CDC prior to detection. A small amount of ozone is detected directly as

$O_3^-$ but the magnitude of this signal is less than 1% of the signal of $CO_3^-$ during ambient sampling. The proposed mechanism of $CO_3^-$ formation is supported by a study using isotopically labelled oxygen to form labelled ozone anions ($^{18}O_3^-$) in a corona discharge source which then reacted with $CO_2$ to form the detected product $C^{18}OO_2^-$ (Ewing and Waltman, 2010). This product supports that a single oxygen is transferred from the ozone anion to carbon dioxide (as in R5a).

    The detection of $NO_2$ proceeds directly through a charge transfer reaction with $O_2(H_2O)_n^-$ to form the detected $NO_2^-$

product (R6). This is expected based upon the high E.A. of $NO_2$ (2.27 eV, (Ervin et al., 1988)) compared to $O_2$ (E.A 0.45 eV).

$$NO_2 + O_2(H_2O)_n^- \rightarrow NO_2^- + O_2 + nH_2O \tag{R6}$$



Oxygen anions are expected to be a highly general reagent ion chemistry, showing sensitivity to an array of analytes. While the focus of this work is on detection of $O_3$ and $NO_2$, detection of hydrogen peroxide, nitric acid, formic acid, sulfur dioxide and other species with the Ox-CIMS has demonstrated good performance (Vermeuel et al., 2019).  An example ambient

mass spectrum recorded at 1 Hz sampling is shown in Fig. 1, with several major peaks highlighted. Also apparent are an abundance of peaks throughout the spectra with high signal intensity. During ambient observations, over one third of masses from $-m/Q$ 27-327 showed signal intensity greater than $1 \times 10^4$ counts per second (cps). A larger survey and classification of oxygen anion reagent ion chemistry to utilize this versatility is underway.

### 2.3 Laboratory calibration

Laboratory calibrations of the Ox-CIMS were performed to determine instrument sensitivity to $O_3$ and $NO_2$. Ozone was generated by passing UHP Zero Air (ZA, Airgas 5.0 grade) through a mercury lamp UV source (Jelight Co, Irvine CA). Outflow from the lamp source was diluted in UHP ZA and split between the Ox-CIMS and a factory calibrated 2B POM ozone monitor (2B Technologies) with an accuracy of $\pm$ 1.5 ppbv, which served as our reference standard. Ozone concentrations were varied over the range 0–80 ppbv and instrument response was determined to generate a calibration curve. $NO_2$ was

delivered from a certified standard cylinder (Scott-Marrin 4.84 $\pm$ 0.1 ppmv). The primary $NO_2$ standard was diluted in UHP ZA to span the range of 0–10 ppbv. Dilutions of calibration standards were made in UHP ZA which was humidified to the desired amount by splitting a portion of the flow through a bubbler containing 18 M$\Omega$ water. $CO_2$ (Airgas Bone Dry grade) was added to the dilution flow to maintain mixing ratios of 380 ppmv for all calibrations (See Section 2.6). A Vaisala HMP 110 sensor continuously measured relative humidity and temperature inline downstream of the Ox-CIMS and POM inlets. All

flows were controlled by mass flow controllers (MKS instruments, 1179C series) with an estimated total uncertainty of 10%. Example calibration curves for $O_3$ and $NO_2$ are shown in Fig. 2. An overview of instrument sensitivity, limits of detection (LOD), and precision to $O_3$ and $NO_2$ is given in Table 1.

### 2.4 Absolute sensitivity

The absolute sensitivity of the Ox-CIMS for detection of analytes is controlled by the kinetics and thermodynamics of the

reagent ion chemistry and the total ion generation and transmission efficiency of the instrument. Under the operational configuration described in Section 2.1, typical reagent ion signal ($O_2^-$ + $O_2(H_2O)_n^-$) ranged from 0.8 to $2.2 \times 10^7$ ions s$^{-1}$ (Fig. S1). The mean total reagent ion signal over 6 weeks of ambient sampling (Section 3.1) was $1.45 \times 10^7$ ions s$^{-1}$. The absolute instrument sensitivity at this reagent ion signal to $O_3$ and $NO_2$ is 180 and 97 ions s$^{-1}$ pptv$^{-1}$ respectively (at 8 g kg$^{-1}$ SH). Total instrument count rate is a complex function of instrument design, instrument ion optics tuning, Po-210 source decay, micro

channel plate (MCP) detector decay, and ToF extraction frequency; all of which are either tunable parameters or vary in time. Conversely, the reagent ion charge transfer or adduct formation chemistry for a given analyte sets a fundamental limit on sensitivity for a given instrument configuration. Sensitivity values can be normalized to a fixed reagent signal count rate ($1 \times 10^6$ ions s$^{-1}$) to isolate the sensitivity component controlled by reagent ion chemistry, separate from total instrument count rate.



Sensitivity values through the remainder of the text are reported as either absolute sensitivities in counts per second (cps pptv$^{-1}$) or normalized sensitivities in normalized counts per second (ncps pptv$^{-1}$). Absolute sensitivity values control instrument limits of detection (LOD) and precision, while normalized sensitivities are used for comparison of calibration factors.

**2.5 Dependence of instrument sensitivity on specific humidity**

The dependence of instrument sensitivity on ambient water content was assessed for specific humidity (SH) ranging between 0–16 g kg$^{-1}$ (approximately 0-80% RH at 25 °C) by triplicate calibrations as shown in Fig. 3. Sensitivity to $O_3$ was seen to have no significant dependence on specific humidity over the range 4-16 g kg$^{-}$. Sensitivity to $NO_2$ has a specific humidity dependence over the range 4–16 g kg$^{-1}$, decreasing from 7.9 to 4.6 ncps pptv$^{-1}$. A 30% and 45% decline in sensitivity was observed from 0 to 4 g kg$^{-1}$ for $O_3$ and $NO_2$ respectively. This low humidity range is rarely sampled in the boundary layer over water surfaces but may be significant in some terrestrial or airborne deployments and would require careful calibration. The SH range from 8 to 16 g kg$^{-1}$ corresponds to approximately 40 to 80% RH at 25 °C which is typical of the humidity range over mid-latitude oceans (Liu et al., 1991). *Ab initio* calculations of $O_2^-(H_2O)_n$ and $O_3^-(H_2O)_n$ clusters performed by Bork et al. (2011) showed that charge transfer from the bare ($n=0$) $O_2^-$ to $O_3$ was exothermic at ca. -160 kJ/mol. At larger cluster sizes of $n = 4$–12, charge transfer becomes less favorable and converges to ca. -110 kJ/mol. An increase in $n$ from 0 to 4 over the SH range 0–4 g kg$^{-1}$ is a potential explanation for the initial decline in sensitivity observed with SH before levelling off from 4–16 g kg$^{-1}$. It is not known if the enthalpy of charge transfer from $O_2^-(H_2O)_n$ to $NO_2$ follows a similar trend with $n$. Ion mobility studies to determine the $O_2^-(H_2O)_n$ cluster size with SH and IMR pressure would provide valuable insight on the observed dependence of sensitivity on water content.

**2.6 Dependence on CO₂**

The ionization pathway for detection of $O_3$ with $O_2^-(H_2O)_n$ reagent ion chemistry differs from typical chemical ionization schemes, in that it involves a two-step reaction of charge transfer to ozone forming $O_3^-$, which then reacts with $CO_2$ to form the detected $CO_3^-$ product (R4-R5). Therefore, we assessed the impact of $CO_2$ mixing ratio in the sample flow on $O_3$ sensitivity as shown in Fig. 4. Calibration curves were generated by diluting ozone in dry UHP $N_2$ and mixing in a flow of variable $CO_2$ (Airgas, Bone Dry Grade) mixing ratios before sampling. At nominally 0 ppmv $CO_2$, the $O_3^-$ ionization product (–48 m/$Q$) was detected with sensitivity of $14 \pm 2$ ncps pptv$^{-1}$ and the $CO_3^-$ product (–60 m/$Q$) at $5 \pm 1$ ncps pptv$^{-1}$. For $CO_2$ mixing ratios from 60 to 500 ppmv, the $O_3^-$ signal is less than 1% of the $CO_3^-$ product and the sensitivity at the $CO_3^-$ product is independent of $CO_2$ within the uncertainty. The presence of a significant fraction (36%) of the $CO_3^-$ product with nominally 0 ppmv $CO_2$ suggests the presence of a slight leak rate of $CO_2$ *via* diffusion through the perfluoroalkoxy alkane (PFA) tubing, or $CO_2$ contamination in the UHP $N_2$ supply. The manufacturer stated upper limit of $CO_2$ in the UHP $N_2$ is 1 ppmv which we take to be the lower limit achievable in our system. A $CO_2$ mixing ratio of only 1 ppmv is still an order of magnitude excess relative to a high end ambient $O_3$ mixing ratio of 100 ppbv. An exponential fit of the $O_3^-$ product *vs* $CO_2$ indicates that $O_3^-$ makes up less than 1% of





the detected ozone at $CO_2$ mixing ratios greater than 10 ppmv. This suggests ambient samples will always have a substantial

excess of $CO_2$ necessary to drive the reaction completely to the $CO_3^-$ product. The measured flat response from 60–500 ppmv

$CO_2$ indicates that natural variability in ambient $CO_2$ will have negligible impact on ambient measurements of ozone. No other

analytes analyzed with the Ox-CIMS show a $CO_2$ mixing ratio dependence, demonstrating $CO_2$ is uniquely involved in this

mechanism and is not a general feature of the oxygen-anion chemistry. All other reported laboratory calibrations reported here

were performed at $CO_2$ mixing ratios of 380 ppmv and all reported sensitivities are for the $CO_3^-$ product. This $CO_2$ dependence

also requires careful consideration during instrument background determinations by UHP $N_2$ overflow which is discussed in

Section 2.8.

**2.7 Dependence on IMR pressure**

Instrument sensitivity to $O_3$ increases with increasing IMR pressure as shown in Fig. 5. The normalized signal of $O_3$ increases

by 175% over the pressure range of 70 to 95 mbar in the IMR when sampling a constant $O_3$ source of 35 ppbv. IMR pressure

was increased in approximately 5 mbar steps, with CDC pressure held constant at 2 mbar, and a three-minute dwell time at

each step to ensure signal and pressure were stabilized. Total reagent ion signal did not change significantly over this pressure

range. Pressures above 95 mbar were not investigated due to concerns over corresponding increases in CDC pressure with the

pinhole and pumping configuration used in this work. There is no evident plateauing in the signal increase over the IMR

pressure range investigated here, indicating that further optimization is likely possible by operating at higher IMR pressures.

The increase in sensitivity with IMR pressure could be fit well with an exponential least squares fit, which is plotted in Fig.5.

The physical meaning of the exponential relationship is not clear. The source of the response of sensitivity to pressure is not

definitive but can possibly be attributed to the increase in the total number of collisions during the 100 ms residence time in

the IMR and the corresponding weakening of those collisions. Higher collisional frequencies also lead to proportionally weaker

collisions which could better preserve a weakly bound $O_2(O_3)(H_2O)_n^-$ cluster and allow a longer lifetime to react with $CO_2$

before dissociation. The operational IMR pressure of 95 mbar used here was empirically selected to maximize sensitivity to

$O_3$ without increasing CDC pressure beyond the desired range. Investigation of higher IMR pressures, up to the operation of

an atmospheric pressure interface, has the potential to further increase the instrument sensitivity to $O_3$.

**2.8 Instrument background and limits of detection**

Instrument backgrounds were assessed by periodically overflowing the inlet with UHP $N_2$ during field sampling. Details of

the inlet and zeroing conditions used are discussed further in Section 3.1. During $N_2$ overflow, $O_3$ displayed a consistently

elevated background on the order of $3.1 \times 10^5$ cps corresponding to $2.1 \times 10^4$ ncps, or approximately 1.3 ppbv $O_3$, at a typical

total reagent ion signal of $1.45 \times 10^7$ cps. A representative background determination is shown in Fig. 6. The magnitude of the

$O_3$ background was observed to vary with the $O_2$:$N_2$ ratio in the reagent ion precursor flow when sampling a UHP ZA overflow

with 380ppm $CO_2$ as shown in Fig. S2. The background $O_3$ count rate was observed to increase from $3.0 \times 10^4$ to $6.3 \times 10^4$

ncps as the $O_2$ volume fraction in the reagent ion delivery gas flow ($f_{O2}$) was increased from 0.05 to 0.4. The dependence of





the background $O_3$ signal on $f_{O_2}$ suggests that the observed background $O_3$ is formed directly in the alpha ion source and is not from off-gassing of inlet and instrument surfaces. The magnitude of this background $O_3$ does not vary when sampling UHP zero air or $N_2$, further confirming that the background $O_3$ is formed directly in the ion source from the $O_2$ used to generate the

reagent ion. An operational $f_{O_2}$ of 0.08 (actual volumetric flow ratio $O_2$:$N_2$ of 200:2200 sccm) was selected to balance maximizing the total reagent ion signal while minimizing the $O_3$ ion-source background (3.1 x $10^5$ cps). The magnitude of this $O_3$ background was observed to be highly consistent during field sampling at a constant $f_{O_2}$ of 0.08 and well resolved from all ambient observations (Fig. S3). The $1\sigma$ deviation of the distribution of normalized adjacent differences of $O_3$ signal during background periods gives an upper limit of variability of 9% between adjacent background periods. A variability of 9%

corresponds to a difference of 70 pptv between subsequent $O_3$ background determinations. The magnitude of this $O_3$ background is a fundamental limit on the achievable limit of detection.

Because $CO_2$ was not added to the UHP $N_2$ overflow during field sampling, the reaction was not driven fully to the $CO_3^-$ product and some $O_3^-$ signal at $-m/Q$ 48 was observed during UHP $N_2$ overflow periods as shown in Fig. S4. The magnitude of the $O_3$ signal observed as $O_3^-$ was approximately 55% of the $CO_3^-$ product (mean 1.2 x $10^4$ and 9.6 x $10^3$ ncps

respectively) during overflow periods. The total sensitivity to $O_3$ as the sum of the $O_3^-$ and $CO_3^-$ was observed to be constant as a function of $CO_2$ as shown in Fig. 4. We therefore assign equal sensitivity to each $O_3$ detection product and took the sum of signal at $O_3^-$ and $CO_3^-$ in order to determine the total background $O_3$ concentration. This issue will be corrected in future deployments by the addition of $CO_2$ to the $N_2$ overflow used for backgrounds which will drive the product fully to $CO_3^-$. The mean background of $O_3$ for the full field sampling period was $1.3 \pm 0.3$ ppbv. The 10 Hz precision of $O_3$ during an individual

$N_2$ overflow period was found to be 0.75%, corresponding to 7.5 pptv as shown in Fig. S5. This suggests that variability in the $O_3$ signal from this background source is constant over short timescales and has a negligible impact on instrument precision during ambient sampling.

The 10 Hz limit of detection for $O_3$ is 42 pptv for a S/N of 3, and a mean background $O_3$ signal of 2.1 x $10^4$ ncps as calculated using Eq. 1, below from Bertram et al., 2011, where $C_f$ is the calibration factor, $[x]$ is the analyte mixing ratio, $t$ is

averaging time in seconds, and $B$ is the background count rate. The optimum LOD from the minimum of the Allan variance at an 11 second averaging time is 4.0 pptv (Fig. S6a).

$$\frac{S}{N} = \frac{C_f[X]t}{\sqrt{C_f[X]t - 2Bt}} \qquad \text{E1}$$

The mean background signal during field sampling for $NO_2$ was 3.5 x $10^3$ ncps which corresponds to 0.28 ppbv. At this background level, the 10 Hz LOD for $NO_2$ is 26 pptv for a S/N of 3. The optimum LOD for $NO_2$ is 2.3 pptv at an averaging

time of 19 seconds, determined from the minimum of the Allan variance (Fig. S6b). The background signal of $NO_2$ is notably above zero indicating either off gassing from inlet walls or a secondary production of $NO_2$ in the instrument. A possible source of this background is from degradation of other species such as nitric acid or alkyl nitrates on the inlet walls. Additional calibration would be necessary to ensure that observed $NO_2$ signal is not a secondary product of other species.





### 2.9 Reagent ion saturation and secondary ion chemistry

During ambient sampling the ozone signal (as $CO_3^-$ detected at –60 m/$Q$) is of comparable magnitude to the $O_2^-$ reagent ion signal as shown in Fig. 1. High analyte concentrations (> 5 ppbv) have been shown previously to result in non-linear calibration curves for unnormalized signals (Bertram et al., 2011; Veres et al., 2008). In our system we do not observe non-linearity in the normalized $O_3$ calibration for our highest concentration calibration point of 80 ppbv despite the $CO_3^-$ signal being larger than the $O_2^-$ reagent ion (9 x $10^6$ cps and 6 x $10^6$ cps respectively). The electron affinity (E.A.) of carbonate is from 3.26 (Hunton

et al., 1985) to >3.34 eV (Snodgrass et al., 1990) and is significantly higher than that of oxygen (E.A. 0.45 eV), making it unlikely that carbonate is involved in charge transfer reactions when excess $O_2^-$ is present. . At high $O_3$ concentrations, the reagent ion signal magnitude is reduced, which necessitates normalizing sensitivities to the 1 x $10^6$ cps of reagent ion signal before quantification. For $NO_2$ (E.A. 2.27 eV), the normalized sensitivity showed no dependence on $O_3$ concentrations from 0 to 80 ppbv. Carbonate reagent ion chemistry has been utilized for detection of $HNO_3$ and $H_2O_2$ *via* adduct formation raising

additional concern about potential secondary ion chemistry (Reiner et al., 1998). In laboratory calibrations, shown in Fig. S7, introduction of 0 to 40 ppb $H_2O_2$ resulted in the titration of the $O_3$ signal of 0.06 ppbv per ppbv $H_2O_2$. $H_2O_2$ was detected as an adduct with $O_2^-$ and not $CO_3^-$ , indicating that $O_2^-$ reagent ion chemistry is more favorable despite high $CO_3^-$ signal intensity. The Ox-CIMS $O_3$ measurement also compared well ($R^2$ =0.99) against an EPA AQS $O_3$ monitor over 1 month of ambient sampling where $H_2O_2$ and $HNO_3$ concentrations both exceeded 5 ppbv at times (see Section 3.1 for further discussion of field

intercomparison), further supporting the $CO_3^-$ detection product as a robust indicator of $O_3$ in complex sampling environments.

*Ab initio* calculations of the binding enthalpies of $O_2^-$ and $CO_3^-$ reagent ions with $H_2O$, $HNO_3$, $H_2O_2$, and $CH_3OOH$ were performed with the MP2/aug-cc-pvdz-PP theory and basis set in order to assess the relative favorability of adduct formation between $O_2^-$ and $CO_3^-$ Adduct formation with $O_2^-$ was favorable relative to $CO_3^-$ by 2.5 to 17 kcal mol$^{-1}$ for all analytes that were calculated. All calculated binding enthalpy values are listed in Table S2.

### 2.10 Reagent ion saturation and secondary ion chemistry

Short term precision of the instrument was assessed by calculating the normalized difference between adjacent 10 Hz data points over a 27-minute sampling period of a constant ambient analyte concentration *via* Eq. 2.

$$NAD = \frac{[X]_n - [X]_{n-1}}{\sqrt{[X]_n [X]_{n-1}}} \hspace{3cm} E2$$

The standard deviation of the Gaussian fit of the distribution of normalized adjacent differences (NAD) is a direct measure of

the short-term instrument precision (Bertram et al., 2011). The 1σ precision from the NAD distribution for 10 Hz sampling of 38 ppbv ozone is 0.74% (Fig. 7). The 10 Hz precision for sampling of 2.3 ppbv $NO_2$ is 1.1% The short-term precision for both analytes was larger than expected if the noise was driven by counting noise alone (10 Hz counting noise limit for $O_3$ and $NO_2$ at the concentrations used above are 0.12% and 0.63% respectively), indicating that other potential points of optimization in the instrument configuration are required to further improve short-term precision. Notably, the observed noise source appears

to be white noise given the Gaussian distribution of the NAD (Thornton et al., 2002b).





Short term precision was assessed as a function of count rate by calculating the NAD for all masses in the spectrum over a stable 27-minute sampling period for both 1 Hz and 10 Hz data averaging. From this assessment, precision was observed to improve approximately linearly in a log-log scaling for count rates between $1 \times 10^3$ and $1 \times 10^6$ cps (Fig. S8). Above $1 \times 10^6$

cps there is an apparent asymptote where precision no longer improves with count rate. For 10 Hz averaging and count rates of $1 \times 10^6$ and $1 \times 10^7$ cps, the corresponding instrument precision is 0.75 and 2% respectively, and appears independent of count rate. The counting noise limited 10 Hz precision for $10^6$ and $10^7$ cps is 0.32% and 0.1% respectively. This precision limit could be driven by an uncharacterized source of white noise in the instrument, including MFC drift, IMR turbulence, ion optic voltage drift, and pump drift. Measurement precision of $O_3$ and $NO_2$ could be improved by a factor of 5 and 2 respectively if

this non-counting noise source of white noise was eliminated.

In theory, detection limits can be improved by signal averaging to a lower time resolution than the 10 Hz save rate. Signal–to–noise ratios are expected to improve with the square root of the integration time. At longer timescales, factors including instrument drift become significant, creating a limit on the upper end of averaging time which optimizes signal–to–noise. This was assessed quantitatively by calculation of the Allan variance as shown in Fig. S6 (Werle et al., 1993).

## 330   3 Field results and discussion

### 3.1 Ozone field calibration and intercomparison

Performance of the Ox-CIMS was compared against a co-located EPA Air Quality System (AQS) $O_3$ monitor (Thermo-Fisher 49i, AQS ID 17-097-1007) over one month of ambient sampling during the Lake Michigan Ozone Study 2017 (LMOS 2017) in Zion, IL (Vermeuel et al., 2019). A regression analysis between the two instruments at one-minute averaging showed strong

agreement ($R^2 = 0.99$) as shown in Fig. 8. Ox-CIMS concentrations were averaged to 1 ppbv bins which was the output data resolution of the EPA data logger system for the (Thermo-Fisher 49i). Error bars are the 1σ standard deviation of each Ox-CIMS bin average. Near one–to–one agreement (slope of 0.99) between instruments lends confidence to the calibration, baselining, and long-term stability of the Ox-CIMS. Background of the Ox-CIMS was determined every 70 minutes by overflowing the inlet with dry UHP $N_2$. Calibration factors for $O_3$ were determined by scaling in-field continuous addition of

a C-13 isotopically labelled formic acid standard to humidity dependent calibration factors for $O_3$ and formic acid determined in lab pre- and post-campaign as described in Vermeuel et al., (2019). The EPA $O_3$ monitor shows a persistent high bias at low $O_3$ concentrations (<10 ppbv) relative to the Ox-CIMS. This discrepancy could arise from known interferences from water, mercury, and other species in 254 nm UV absorbance detection of ozone (Kleindienst et al., 1993).

### 3.2 Eddy covariance experiment overview

The Ox-CIMS was deployed to the 330 m long Ellen Browning Scripps Memorial Pier (hereon referred to as Scripps Pier) at the Scripps Institute of Oceanography (32° 52.0' N, 117° 15.4' W) during July and August 2018 for EC measurements of $O_3$





vertical fluxes. This site has been used regularly for EC flux observations from our group and others (Ikawa and Oechel, 2015; Kim et al., 2014; Porter et al., 2018). The Ox-CIMS was housed in a temperature-controlled trailer at the end of the pier. The Ox-CIMS sampled from a 20 m long PFA inlet manifold with the intake point co-located with a Gil-Sonic HS-50 sonic

anemometer which recorded 3-dimensional winds sampling at 10 Hz. The Ox-CIMS inlet and sonic anemometer were mounted on a 6.1 m long boom that extended beyond the end of the pier to minimize flow distortions. The inlet height was 13 m above the mean lower low tide level. The Ox-CIMS inlet was located 8 cm below the sonic anemometer with a 0 cm horizontal displacement. The inlet manifold consisted of a 0.64 cm i.d. sampling line, a 0.64 cm i.d. overflow line, and a 0.47 cm i.d. calibration line all made of PFA. The inlet sample line was pumped at 18-23 slpm (Reynolds number 3860-4940) by a dry

scroll pump (SH-110, Agilent) to ensure a fast time response and maintain turbulent flow. Flow rates in the inlet sample line were recorded by a mass flow meter but were not actively controlled. The inlet manifold, including calibration and overflow lines, was held at 40 °C *via* a single resistively coupled circuit along the length of the manifold and controlled by a PID controller (Omega, model CNi 16). The Ox-CIMS front block and IMR were held at 35 °C. The Ox-CIMS subsampled 1.5 slpm from this inlet manifold through a critical orifice into the IMR. Ambient humidity and temperature were also recorded

in-line downstream of the subsampling point.

### 3.2.1 Calibration

Instrument sensitivity was assessed by the standard addition of a C–13 isotopically labelled formic acid standard for 3 minutes every 35 minutes at the ambient end of the inlet manifold. Ozone mixing ratios were determined by scaling the humidity dependent sensitivity of $O_3$ from pre- and post-campaign calibrations to the field calibrations of C–13 formic acid. Ambient

$O_3$ was also measured at 10 s time-resolution with a 2B technologies Personal Ozone Monitor (POM). The POM had a separate 10 m long, 0.47 cm i.d. PFA sampling line located 12 m from the Ox-CIMS inlet manifold and sonic anemometer. The POM was used as an independent verification of the Ox-CIMS measurement and was not used for calibration.

### 3.2.2 Backgrounds and inlet residence time

Instrument backgrounds were determined every 35 minutes by overflowing the entire inlet manifold with dry UHP $N_2$.

Background and ambient count rates were first converted to concentrations using the laboratory determined humidity dependent sensitivities for $O_3$ and $NO_2$ scaled to the C-13 formic acid standard addition sensitivity. Background concentrations of $O_3$ and $NO_2$ from before and after each 30-minute ambient sampling period were interpolated over the ambient sampling period which was then subtracted from each 10 Hz concentration data point to obtain a background corrected time-series. Background concentrations of $O_3$ had a mean 1.5 ppbv and a drift of 1% between adjacent background periods, determined by

the distribution of the NAD of the mean background concentrations.

The signal response during dry $N_2$ overflows were fit to a bi-exponential decay function to characterize inlet gas evacuation time ($\tau_1$) and wall interaction times ($\tau_2$) (Ellis et al., 2010). Best fit estimates for $\tau_1$ of $O_3$ and $NO_2$ across overflow periods were from 0.7 to 1.2 seconds and accounted for more than 80% of the decay. For $O_3$, $\tau_2$ was found to be negligible at


less than 0.3 s indicating $O_3$ has minimal equilibration with the inlet walls. $\tau_2$ for $NO_2$ was determined to be longer at 3.2 s.

This suggests a potential interference at the $NO_2$ peak, as $NO_2$ is expected to have minimal equilibration, similar to $O_3$. $NO_2$ also shows a continually elevated signal during overflow periods suggesting off gassing from inlet or instrument surfaces. The cause of this slow $NO_2$ decay and elevated background is not clear but could be from degradation of nitric acid or nitrate containing aerosol on the instrument surfaces.

The instrument response time ($\tau_r$) for $O_3$ can be calculated during zeroing periods as the time required for the signal

to fall to 1/e of its initial value. The response time of the instrument was calculated for each overflow period during field sampling, with a mean value of 0.9 s. The cutoff frequency ($f_{cut}$) of the instrument is defined as the frequency where the signal is attenuated by a factor of $1/\sqrt{2}$ (Bariteau et al., 2010). The cutoff frequency can also be calculated from $\tau_r$ according to Eq. 3.

$$f_{cut} = \frac{1}{2\pi\tau_r} \hspace{6cm} \text{E3}$$

The calculated $f_{cut}$ from the measured mean response time was 0.18 Hz. This value suggests that minimal attenuation in the flux signal (cospectra) should be apparent at frequencies less than 0.18 Hz. The instrument response time and thus cutoff frequency are function of the flow rate and sampling line volume. The flow rate of 18-23 slpm was the maximum achievable with the tubing and pumping configuration used here but could be improved in future to minimize tubing interactions and shift $f_{cut}$ towards higher frequencies.

**3.2.3 Eddy covariance flux method**

The transfer of trace gases across the air−sea interface is a complex function of both atmospheric and oceanic processes, where gas exchange is controlled by turbulence in the atmospheric and water boundary layers, molecular diffusion in the interfacial regions surrounding the air−water interface, and the solubility and chemical reactivity of the gas in the molecular sublayer. The flux ($F$) of trace gas across the interface is described by Eq. 4, as a function of both the gas-phase ($C_g$) and liquid phase

($C_l$) concentrations and the dimensionless gas over liquid Henry's law constant ($K_H$), where $K_t$, the total transfer velocity for the gas (with units cm s$^{-1}$), encompasses all of the chemical and physical processes that govern air−sea gas exchange. Surface chemical reactivity terms to the gas exchange rate are incorporated into the $K_t$ term.

$$F = -K_t\left(C_g - K_H C_l\right) \hspace{5cm} \text{E4}$$

Trace gas flux ($F$) can be measured with the well-established eddy covariance (EC) technique where flux is defined

as the time average of the instantaneous covariances from the mean of vertical wind ($w$) and the scalar magnitude (here $O_3$) shown in Eq. 5. Overbars are means and primes are the instantaneous variance from the mean. Here $N$ is the total number of 10 Hz data points during the 27-minute flux averaging period.

$$F = \frac{1}{N}\sum_{i=1}^{N}(w_i - \overline{w})\left(O_{3,i} - \overline{O_3}\right) = \langle w'O_3' \rangle \hspace{3cm} \text{E5}$$

$$v_d = \frac{F}{C_g} \hspace{6cm} \text{E6}$$


For purely depositing species where the water side concentration is negligible, $C_l$ and $K_H$ can be neglected in Eq. 4 and $K_t$ can be reformulated into a deposition velocity ($v_d$) calculated according to Eq. 6, where $\overline{C_g}$ is the mean gas phase mixing ratio during the flux averaging period. A summary of concentration and flux results for the full deployment period are given in Table 2.

**3.3 General Data Corrections**

Several standard eddy covariance data filters and quality control checks were applied before analysis. General filters included:

1.) Wind sector: Only periods of mean onshore flow (true wind direction 200-360°) were used.
2.) Friction velocity: A friction velocity ($U_*$) threshold of 0.1 m s$^{-1}$ was applied to reject periods of low shear driven turbulence (Barr et al., 2013).
3.) Stationarity: Each 27-minute flux period was divided into five even non-overlapping subperiods. Flux periods were
rejected if any of the subperiods differed by more than 40% (Foken and Wichura, 1996).

The selected $U_*$ threshold of 0.1 m s$^{-1}$ is lower than the 0.2 m s$^{-1}$ used frequently as a default but is consistent with other marine flux studies where surface roughness lengths are significantly smaller than over terrestrial surfaces (Porter et al., 2018). Methods of determining site specific $U_*$ thresholds typically require long-term data series which were not available here (Papale et al., 2006). Papale et al., 2006, applied a minimum $U_*$ threshold of 0.1 m s$^{-1}$ for forest sites and 0.01 m s$^{-1}$ for short
vegetation sites where typical $U_*$ values are lower. A default $U_*$ of 0.2 m s$^{-1}$, which is a common value selected for terrestrial environments would reject over 80% of flux periods from our data set. The selected $U_*$ filter results in 57% of the wind direction filtered flux periods being rejected. Outliers in $v_d(O_3)$ and the flux limit of detection were determined and removed for points three scaled median absolute deviations from the median. This outlier filter removed an additional 11 data points. After all quality control filters were applied, 84% of flux periods were rejected leaving 157 quality-controlled flux periods.
Eddy covariance flux values were calculated using 27-minute time windows. The $O_3$ timeseries was detrended with a linear function prior to the flux calculation. The $O_3$ and vertical wind data were despiked using a mean absolute deviation filter before the eddy covariance flux calculation following Mauder et al., (2013).

**3.3.1 Planar Fit Wind Coordinate Rotation**

Coordinate rotation of the $u$, $v$, and $w$ wind components was performed by the planar fit method to remove unintentional tilts
in the sonic mounting and account for local flow distortions (Wilczak et al., 2001). Briefly, the mean $u$, $v$, and $w$ wind components and the stress tensor were determined for each 15-minute onshore flow period during the full campaign. A linear regression was used to find the best fit of a plane with a coordinate system where the z-axis is perpendicular to the mean streamline. Individual 27-minute flux periods are then rotated such that the x-axis is along the mean wind and $\bar{v} = 0$. Vertical wind velocity ($\bar{w}$) in any individual rotation period may be non-zero due to mesoscale motions but $\bar{w}$ for the full campaign is



zero. The residual mean vertical velocity in any individual rotation period is subtracted out, so it does not impact the Reynolds averaging.

### 3.3.2 Lag time shift

The Ox-CIMS signal is delayed relative to the sonic anemometer due to transit time in the inlet line which must be accounted for before calculating the covariance between the vertical wind and analyte concentration. The cross-covariance of the two

timeseries were first calculated within a ± 5 s window to determine the lag time of the Ox-CIMS and synchronize with the anemometer. The volumetric evacuation time of the inlet is 1.65 to 2.1 s for the inlet volume and flow rates of 18 to 23 slpm used in this study. Following the method and terminology outlined in Langford et al. (2015), the position of the maximum (MAX) of the cross-covariance is taken as the lag time needed to align the vertical wind and analyte concentration for that flux period. A representative lag time determination with a larger lag window (± 10 s) using the MAX method is shown in Fig. 10.

In low signal-to-noise (SNR) data, the use of the MAX leads to high variability in the determined lag time caused by uncertainty in the position of the peak in the cross-covariance. This results in a systematic high bias on the absolute magnitude of the resulting flux. The position of the maximum of a centered running median (AVG) function of the cross-covariance is an alternative method to determine lag time with less expected bias for low SNR data (Langford et al., 2015; Taipale et al., 2010). Lag times for each $O_3$ flux period determined by the MAX and a 10 point AVG method showed reasonable agreement, with a

campaign average lag time from the MAX with a mean of 1.0 seconds and the AVG at 0.7 seconds (Fig. S9). This agreement suggests that a clear peak in the cross-covariance was present for most flux periods leading to a convergence of the two methods. This lag time also shows agreement with the inlet response time of 0.9 s determined during dry $N_2$ overflows. Due to the convergence of the determined lag times around a central value, a prescribed lag time of 0.9 s was used for all reported $v_d(O_3)$ values. A prescribed lag time has the least bias to extreme values caused by noise, provided that the true lag time is

known well (Langford et al., 2015). Deposition velocities were then recalculated with the prescribed lag time of 0.9 s and with the MAX and AVG method over a narrower lag window of ± 3 which is expected to be physically reasonable range for the flow rate and inlet line volume. The mean $v_d(O_3)$ using the prescribed, MAX, and AVG lag times were 0.011, 0.010, and 0.010 cm s$^{-1}$ respectively, suggesting the campaign mean value was relatively insensitive to the lag time method.

### 3.4 Cospectra and Ogives

The frequency weighted cospectrum of $O_3$' with $w$' has a well characterized form with exhibited dependence on wind-speed and measuring height (Kaimal et al., 1972). Comparison of observed cospectra shape against the idealized Kaimal cospectra is useful to validate that the observed signal was not significantly attenuated at low or high frequencies. Cospectral averaging is performed by binning frequency into 50 evenly log spaced bins and normalizing the integrated cospectra to 1. The integral of the unnormalized cospectra is the flux for that observation period. The mean wind-speed binned cospectra of sensible heat

and $O_3$ appear to match well with the idealized Kaimal cospectra for an unstable boundary layer at sampling height $z = 13$m (Fig.11).





The ogive is the normalized cumulative distribution of the cospectra, which is used to validate both that no high-frequency attenuation is present and that the flux averaging time is sufficiently long that all frequencies contributing for the flux is captured. Figure 11 shows the averaged cospectra and ogives for $O_3$ and sensible heat flux from the average of two flux averaging periods 14:10 – 15:20 on July 20[th]. The asymptote to 1 at low frequencies validates that the 27-minute flux averaging time was sufficiently long for this site to capture the largest flux carrying eddies. High pumping rates in sampling line ensured that turbulent flow was always maintained in the line (Reynolds number 3860-4940). Higher Reynolds numbers in the turbulent regime lead to smaller high frequency attenuation (Massman, 1991). The overlap of the idealized Kaimal curve and the observed sensible heat and $O_3$ ogives suggest that high frequency attenuation in the sampling line is minimal above 0.2 Hz, consistent with our calculated $f_{cut}$ of 0.18 Hz. Due to the small magnitude of the $O_3$ EC flux there is low signal to noise in the cospectra at high frequency for many of the flux averaging periods. This makes application of cospectra based correction factors challenging and likely to introduce added variance on the signal. We calculate the high frequency correction transfer function for turbulent attenuation in a tube from Massman, (1991) as a constraint, which is shown in Fig. 11b. This transfer function shows attenuation primarily above 1 Hz and is not sufficient to describe the observed attenuation above 0.2 Hz. This implies that the attenuation observed cannot be explained only as turbulent smearing in the inlet and that other wall interactions are likely present. We also calculate the attenuated flux from the model of Horst, (1997) shown in Eq. 7, for a response time ($\tau_c$) of 0.9 s, and a wind speed of 3 m s$^{-1}$ to be 13%.

$$\frac{F_m}{F_x} = \frac{1}{1+(2\pi n_m \tau_c U/z)^\alpha}$$
E7

Where $F_m/F_x$ is the ratio of the measured flux to the unattenuated flux, $U$ is wind speed, $z$ is measurement height, and $n_m$ and $\alpha$ are scaling factors for an unstable boundary layer taken as 0.085 and 7/8 respectively. This flux attenuation of 13% is in reasonable agreement with the attenuation visible in the ogives shown. Given the uncertainty in the cospectra at high frequencies it is difficult to make a quantitative statement as to whether the observed attenuation is consistent throughout the measurement period. We therefore elect to report the uncorrected flux value here while acknowledging they are potentially a lower estimate with reasonable correction factors on the order of 15%.

### 3.5 Uncertainty and flux limit of detection

Variance in the atmospheric $O_3$ signal was estimated by calculating the autocovariance of the signal during a 27-minute flux averaging period (Fig. S11). Uncorrelated white noise only contributes to the first point in the autocovariance spectrum, while autocovariance at longer time shifts represents real atmospheric variance or correlated instrument drift (Blomquist et al., 2010; Langford et al., 2015). For the analysed period, white noise is typically 45 to 65% of the total variance and atmospheric variance is 35 to 55%. This corresponds to a standard deviation from white noise $\sigma_{O_3, noise}$ of 0.4 ppbv.





The error in each flux averaging period ($LOD_\sigma$) can be determined by taking the standard deviation of the cross-covariance between vertical wind speed and mass spectrometer signal at lag times significantly longer than the calculated true lag time

(Spirig et al., 2005; Wienhold et al., 1995). The random flux error is determined using lag windows of -150 to -180 and 150 to 180 s, which are significantly larger than the true lag time from sensor separation of 0.9 s as shown in Figure S12. The selection of the -150 to -180 and 150 to 180 s lag windows is somewhat arbitrary and may still capture organized atmospheric structure that persists over long time periods. We also calculate the root mean squared deviation ($LOD_{RMSE}$) of the cross-covariance over the same lag windows as proposed by Langford et al., (2015), which captures the variance in the cross-

covariance in those regions but also accounts for long term offsets from zero in the cross-covariance. The resulting error from the $LOD_\sigma$ and $LOD_{RMSE}$ methods showed good correlation (Fig S13), with periods where the $LOD_{RMSE}$ error is larger. We apply the RMSE method for our reported flux error determination. The final deposition velocity limit of detection was determined for each 27-minute flux averaging period by multiplying the $LOD_{RMSE}$ error by 1.96 to give the flux limit-of-detection at the 95% confidence level. The flux error was then divided by the mean $O_3$ concentration for that averaging period

to convert from flux to deposition velocity units. The campaign ensemble flux $LOD_{RMSE}$ was 0.0042 cm s$^{-1}$, calculated using Eq. 8 following Langford et al.,(2015). A total of 62 out of 157 (39%) flux periods had deposition velocities below the campaign ensemble LOD. These values are still included in the reported mean $v_d(O_3)$.

$$\overline{LOD} = \frac{1}{N}\sum_{i=1}^{N} LOD^2 \hspace{6cm} E8$$

### 3.6 Density fluctuation corrections

The Ox-CIMS measures $O_3$ as the apparent mixing ratio relative to moist air, which means fluctuations in the density of air due to changes in temperature, pressure, and humidity could introduce a bias in the EC flux measurement (Webb et al., 1980). The temperature and pressure in the Ox-CIMS and sampling lines were both actively controlled during sampling, making density fluctuations from those sources negligible. The long (20 m) inlet sampling line used likely also dampened a substantial portion of the water vapor flux. However, without a direct measure of water vapor fluctuations collocated with the Ox-CIMS

this is difficult to assess directly. We therefore calculate a conservative estimate of this correction factor from Eq 45b. in Webb et al., (1980), assuming a latent heat flux of 50 W m$^{-2}$ and neglecting the sensible heat term which is removed by active heating of the inlet which is removed by active heating of the inlet. For a specific humidity of 12 g kg$^{-1}$, a temperature of 293 K, a pressure of 1 atm, and an $O_3$ mixing ratio of 40 ppbv; we calculate a flux correction term of 2.6 x 10$^9$ molecules cm$^{-2}$ s$^{-1}$, which is 24% of our mean measured flux of -1.1 x 10$^{10}$ molecules cm$^{-2}$ s$^{-1}$. We expect that the actual density correction for our

instrument much smaller given that water vapor fluctuations were likely dampened in the inlet line, and the high latent heat flux used in the calculation (50 W m$^{-2}$). Due to the uncertainty in this correction term for our instrument, we do not add it to our measured flux values and instead use the calculated value above as a conservative constraint on the magnitude. The addition of a Nafion drier on the inlet has been successfully implemented in other $O_3$ flux instruments to fully remove water fluctuations and will be used in future deployments of the Ox-CIMS (Bariteau et al., 2010).





### 3.7 Flux divergence

#### 3.7.1 Surface NO emissions

The observed dry deposition velocity of ozone is potentially biased by simultaneous air-sea exchange of nitric oxide (NO). NO is expected to be emitted from the ocean on the order of 1 x $10^8$ molecules cm$^{-2}$ s$^{-1}$ with dependence on dissolved surface nitrate and solar irradiance (Zafiriou and McFarland, 1981). This NO source near the surface will cause titration of $O_3$ to $NO_2$ resulting in a positive bias for the observed $v_d(O_3)$. Assuming a maximum NO emission flux of 5 x $10^8$ molecules cm$^{-2}$ s$^{-1}$ and that all NO reacts with $O_3$ before being advected to the sensor height, the resulting $O_3$ flux bias would be -5 x $10^8$ molecules cm$^{-2}$ s$^{-1}$. Our mean case of 40 ppbv $O_3$ and $v_d (O_3)$ of 0.011 cm s$^{-1}$ corresponds to a flux of -1.1 x $10^{10}$ molecules cm$^{-2}$ s$^{-1}$. Therefore, the resulting bias in observed $v_d(O_3)$ from NO emissions is 4.5% or 4.9 x $10^{-4}$ cm s$^{-1}$. This value is an upper limit for expected ocean NO emissions and is well within the uncertainty of the observed $v_d(O_3)$.

#### 3.7.2 Free troposphere entrainment

The entrainment of $O_3$ enhanced or depleted air in the free troposphere to the marine boundary layer (MBL) creates a potential flux gradient that will contribute to the measured flux values at the near surface measurement height ($z_o$) of 13 m. The magnitude of this flux gradient depends on the magnitude of the $O_3$ concentration gradient ($\Delta C$) and the entrainment velocity ($w_e$) of air from free troposphere into the MBL. Faloona et al. (2005), reported entrainment velocities from 0.12 to 0.72 cm s$^{-1}$ and an enhancement in $O_3$ ($\Delta C$) of 20 ppbv in the free troposphere relative to the boundary layer in the summertime eastern subtropical pacific. Using those values and Equations 9 and 10 below we calculate the percent fractional error from entrainment on the observed flux for a range of reasonable $\Delta C$ and $w_e$ as shown in Fig. 12 (Blomquist et al., 2010).

$$\frac{\Delta F_{0,est}}{F_0} = \frac{z}{z_i}\left(\frac{F_i}{F_0} - 1\right) \qquad \text{E9}$$

$$F_i = w_e \Delta C \qquad \text{E10}$$

Where $z$ is the boundary layer height, $z_i$ is the measurement height, and $F_i$ and $F_0$ are the entrainment flux and surface flux respectively. We use the SIO measurement height ($z_o$) = 13 m and mean surface flux ($F_o$) = -4.4 x $10^{-3}$ ppbv m s$^{-1}$ (from $v_d$ = 0.011 cm s$^{-1}$ and [$O_3$] = 40 ppbv ), and an $O_3$ mixing ratio gradient ($\Delta C$) from -20 to +20 ppbv in the free troposphere relative to the boundary layer. The resulting fractional error in our observed mean surface flux from Scripps Pier using the values from Faloona et al, 2005 ($\Delta C$ of +20 ppbv, MBL height of 800m) is 6.25% for $w_e$ of 0.12 and 44% for 0.72 cm s$^{-1}$. This entrainment flux error is clearly significant for marine $O_3$ flux measurements assuming there is a gradient of $O_3$ in the free troposphere relative to boundary layer. This entrainment flux error is independent of the surface flux instrument measurement error and adds a systematic bias on the surface flux measurement. This calculation also makes clear that marine $O_3$ measurements should be made as close to the surface as possible, and that the $O_3$ concentration gradient and entrainment rate should be explicitly measured if possible. We do not have an explicit measure of $\Delta C$, $w_e$, or the MBL height so we tentatively assign entrainment error of up to 44% from the maximum values of those parameters reported in Faloona et al. (2005). We emphasize this source





of uncertainty is independent of the $O_3$ sensor and is a systematic bias that should be considered in all $O_3$ air-sea exchange determinations.

**4 Fast $NO_2$ measurements, eddy covariance and $O_3$ titration**

Discussion of EC flux results have been limited to $O_3$ because ocean—atmosphere exchange of $NO_2$ is expected to be small
and below the limit of detection of our instrument. The potential flux divergence from the reaction of $O_3$ with NO is also below the instrument flux limit of detection as discussed in section 3.6. However, over terrestrial surfaces where $NO_2$ emissions can be large, we expect this instrument would be well suited for measuring $NO_2$ flux. Following Equation 1 in Bariteau et al., (2010), we calculate an expected flux LOD of $4.3 \times 10^9$ molecules cm $s^{-2}$ $s^{-1}$ (1.6 pptv m $s^{-1}$) for an $NO_2$ mixing ratio of 1 ppbv and a friction velocity of 0.2 m $s^{-1}$.
Observations of a short duration NO plume from a boat motor starting near our inlet at Scripps Pier highlights the utility of the simultaneous $O_3$ and $NO_2$ detection from this instrument (Fig. 13). Highly localized NO emissions were observed as the titration of $O_3$ and prompt production of $NO_2$. Observed total odd oxygen ($O_x = O_3 + NO_2$) was conserved during this titration event, where $NO_2$ and $O_3$ concentrations were determined from independent calibration factors and backgrounds. The 1:1 conversion of $O_x$ from $O_3$ to $NO_2$ shown in Fig. 13b, validates the laboratory generated instrument calibration factors for
$O_3$ and $NO_2$. The temporal agreement of the $O_3$ and $NO_2$ signals also demonstrates that both $O_3$ and $NO_2$ are transmitted through the inlet and detected with nearly identical instrument response times. This simultaneous detection of both $O_x$ species is likely also well suited for mobile sampling in the presence of dynamic NO emission sources, which challenge other fast ozone measurements. This method would also be well suited for direct measurement of flux divergence in the presence of strong surface NO emission sources.

**5 Conclusions and Outlook**

This study demonstrated the utility of oxygen anion chemical ionization mass spectrometry for the fast and sensitive detection of $O_3$ and $NO_2$. Field measurements of $O_3$ dry deposition to the ocean surface from Scripps Pier, La Jolla CA demonstrate that this method has suitable time response, precision, and stability for successful EC measurements. The mean measured $v_d(O_3)$ with the Ox-CIMS is in good agreement with prior studies of $O_3$ ocean-atmosphere exchange. Further optimization and
characterization of the Ox-CIMS is ongoing, including efforts to validation the specificity of the $NO_2$ detection, addition of a Nafion drier system, and better background determination methods. While this work has focused primarily on the deposition of $O_3$ to the ocean surface, the demonstrated instrument performance suggests the Ox-CIMS to be highly capable of $O_3$ and $NO_2$ flux measurements in the terrestrial biosphere and urban environments and from mobile platforms.



**Author Contributions**

GAN, MPV, and THB designed the lab and field experiments and GAN and MPV collected all data. GAN lead the data processing, interpretation and analysis with MPV contributing. GAN prepared the manuscript with contributions from all co-authors. THB supervised all work and contributed to data analysis, writing and editing of the manuscript.

**Acknowledgments**

This work was supported by National Science Foundation (NSF) Grant GEO AGS 1829667.

The authors thank the staff at Scripps Pier, Scripps Institute of Oceanography and at the UW—Madison Department of Limnology for support in instrument deployments.

Glenn M. Wolfe is gratefully acknowledged for publicly providing a Matlab based "FluxToolbox" of analysis scripts, portions of which were altered for use in this analysis. Code is archived at https://github.com/AirChem

This research was performed using the computing resources and assistance of the UW-Madison Center for High Throughput

Computing (CHTC) in the Department of Computer Sciences. The CHTC is supported by UW-Madison, the Advanced Computing Initiative, the Wisconsin Alumni Research Foundation, the Wisconsin Institutes for Discovery, and the National Science Foundation, and is an active member of the Open Science Grid, which is supported by the National Science Foundation and the U.S. Department of Energy's Office of Science.

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


| Species | Normalized Sensitivity (8 g kg⁻¹ SH, 1σ) | Absolute Sensitivity (8 g kg⁻¹ SH) | LOD optimum | LOD (1 Hz) | LOD (10 Hz) | Background (cps, 1σ) | Precision (10 Hz) | Field Calibration $R^2$ |
|---|---|---|---|---|---|---|---|---|
| $O_3$ | $12.4 \pm 1.2$ ncps pptv⁻¹ | 180 cps pptv⁻¹ | 4.0 pptv (11s) | 13 pptv | 42 pptv | $3.1 \times 10^5$ $\pm 5.0 \times 10^4$ | 0.74% | 0.99 |
| $NO_2$ | $6.7 \pm 1.0$ ncps pptv⁻¹ | 97 cps pptv⁻¹ | 2.3 pptv (19s) | 9.9 pptv | 32 pptv | $5.1 \times 10^4$ $\pm 1 \times 10^4$ | 1.1% | — — — |

**Table 1. Summary of instrument sensitivity, precision, and accuracy for detection of O₃ and NO₂ from laboratory calibrations. Sensitivity is reported at a specific humidity (SH) of 8 g kg⁻¹ which corresponds to 40% RH at 25 °C. All limits of detection (LOD) are for a S/N = 3. The optimum LOD is reported as the LOD at the optimum averaging time determined by the minimum of the Allan variance spectrum. Optimum averaging times were determined to be 11 s for O₃ and 19 s for NO₂. The reported field comparison ($R^2$) is from a regression of 1-minute bin averaged ozone concentration from the Ox-CIMS with an EPA (Thermo-Fisher 49i) monitor in Zion, Il during four weeks of ambient observation shown in Fig. 7.**



| Species | Concentration Mean (ppb) & 1σ range | 5th to 95th percentile Concentration Range (ppbv) | $v_d$ mean (cm s⁻¹) | $v_d$ 20-80% range (cm s⁻¹) | $v_d$ LOD 1.96σ (cm s⁻¹) |
|---|---|---|---|---|---|
| $O_3$ | 38.9 ± 12.3 | 16.9 – 56.9 | 0.011 | -0.0048 – 0.0249 | 0.0042 |
| $NO_2$ | 4.7 ± 5.5 | 0.45-16.9 | ---------- | ---------- | ---------- |

**Table 2. Overview of flux and concentration measurements of $O_3$ and $NO_2$ from Scripps Pier. Concentration ranges are reported for all periods of onshore winds. Flux results are reported only for final quality-controlled flux periods Ozone mean deposition velocity ($v_d$) was well resolved from the campaign ensemble average LOD of 0.0042 cm s⁻¹. Reported $v_d$ LOD is the ensemble mean of the LOD determined by the RMSE method at long lag times for each 27-minute flux period. 39% of quality-controlled flux periods fell below the campaign ensemble LOD. Deposition velocity of $NO_2$ across the air-sea interface is expected to be small (<0.002 cm s⁻¹) and was consistently below the LOD of our instrument so no values are reported here.**






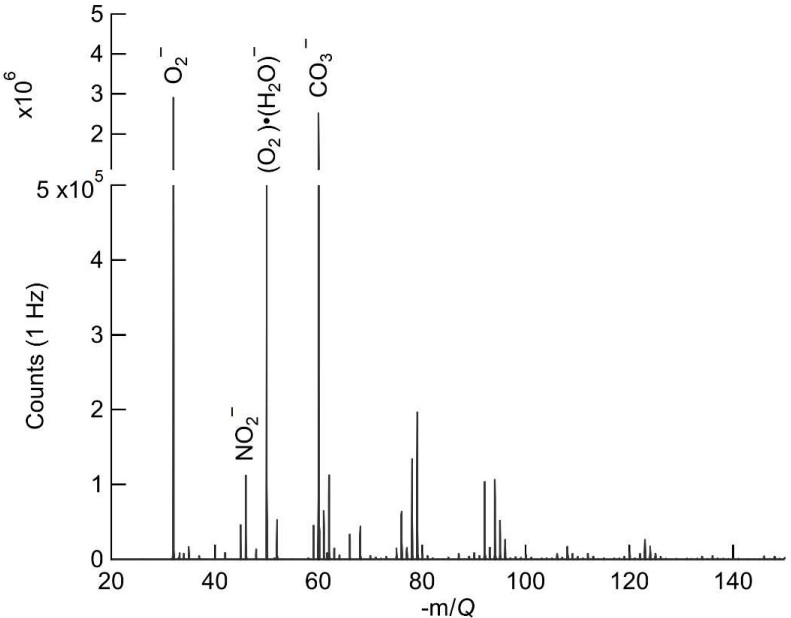

**Figure 1: Ox-CIMS mass spectra collected at 1 Hz and mass resolution of 950 *M/ΔM* (at −60 m/*Q*), with major peaks highlighted.**
**$O_2^-$ and $O_2(H_2O)^-$ at −32 m/*Q* and −50 m/*Q* respectively are the two observed forms of the reagent ion. The detected ozone product**
**($CO_3^-$, −60 m/Q) is of comparable magnitude to the $O_2^-$ reagent ion during ambient sampling. $NO_2$ is detected as the charge transfer**
**product $NO_2^-$ at −46 m/*Q*. Masses greater than −150 m/*Q* contribute less than 2% to the total signal and are not plotted.**






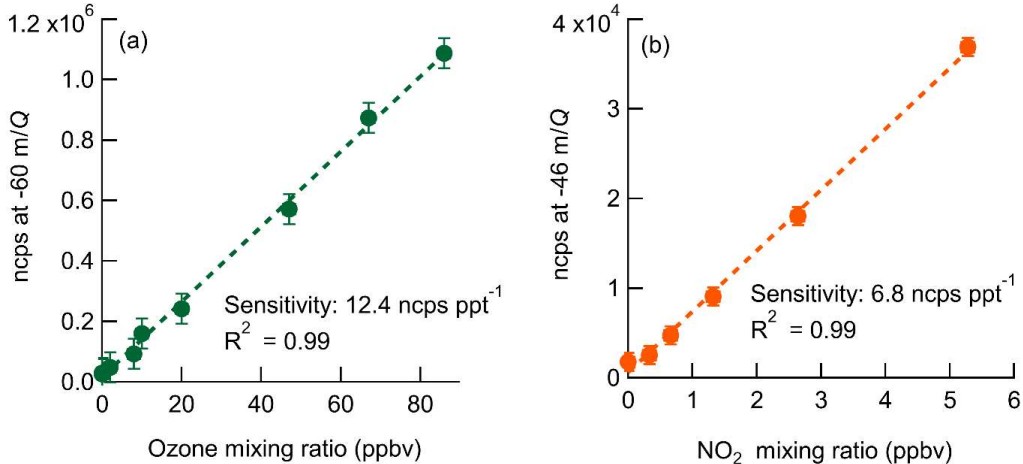


**Figure 2: Normalized calibration curves of O₃ (a) and NO₂ (right) at 8 g kg⁻¹ specific humidity (approximately 40% RH at 25 °C). Ozone is detected as CO₃⁻ at −60 m/Q. NO₂ is detected as the charge transfer product (NO₂⁻) at −46 m/Q. Error bars are the standard deviation in normalized count rate for each measurement point.**



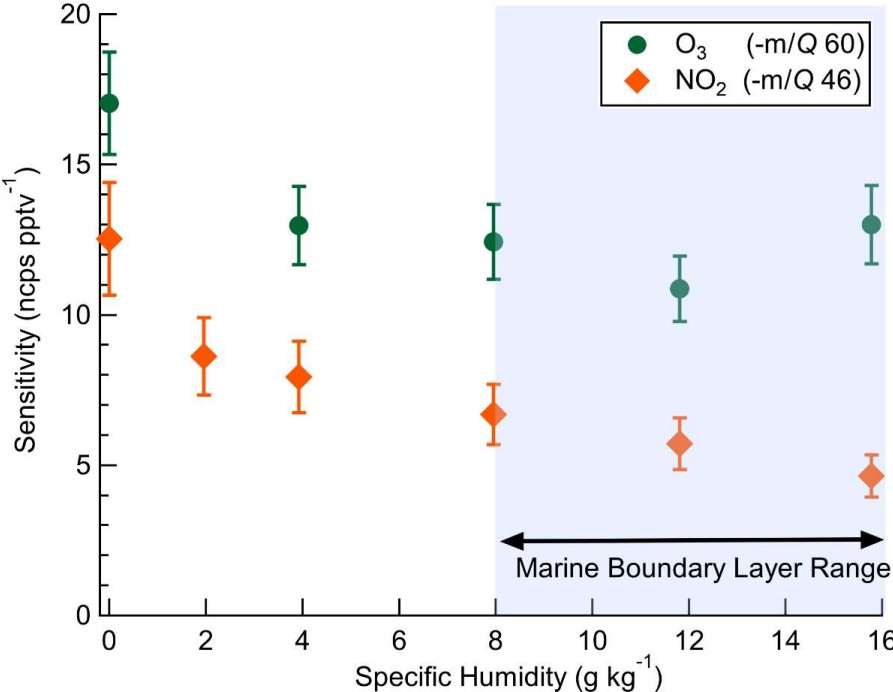


**Figure 3: Dependence of O₃ and NO₂ sensitivities on specific humidity. Error bars indicate standard deviation of triplicate calibration curves. The blue shaded region from SH 8–16 g kg⁻¹ is the approximate typical range of specific humidity in the mid-latitude marine boundary layer.**





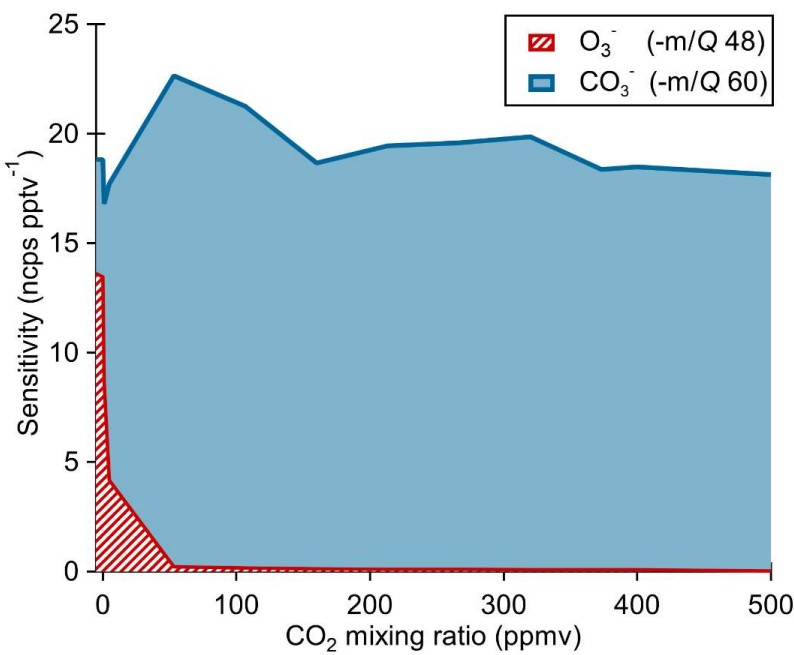


**Figure 4: Ox-CIMS cumulative sensitivity to O₃ detected either directly as $O_3^-$ or as $CO_3^-$ as a function of CO₂ mixing ratio. The sum of sensitivity as $O_3^-$ and $CO_3^-$ shows that total sensitivity to O₃ is conserved as the product distribution shifts with CO₂ mixing ratio. Greater than 99% of O3 is observed as $CO_3^-$ at CO₂ mixing ratios greater than 60 ppmv.**



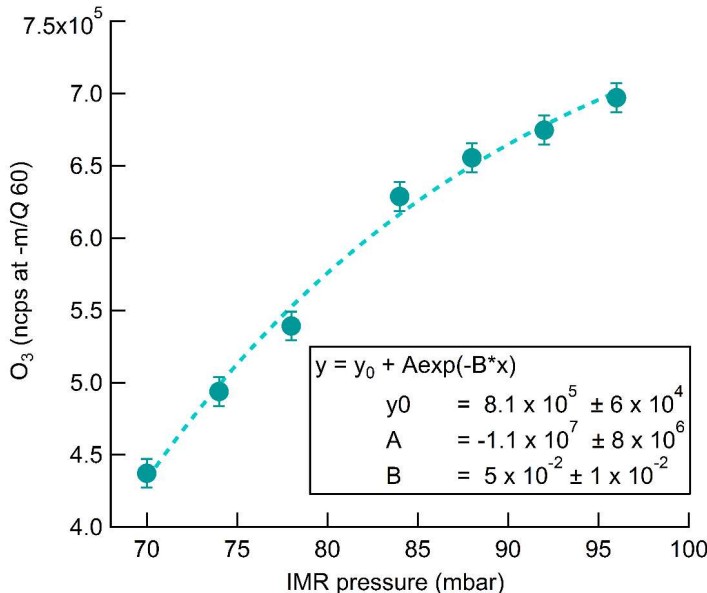

**Figure 5. Normalized count rate of $CO_3^-$ (–60 m/Q) ozone detection product as a function of pressure in the IMR during sampling of a constant 35 ppbv $O_3$ source. The exponential fit of the data is shown by the dashed line. Fit parameters are included to allow for calculation of potential sensitivity improvements with further increase in IMR pressure.**





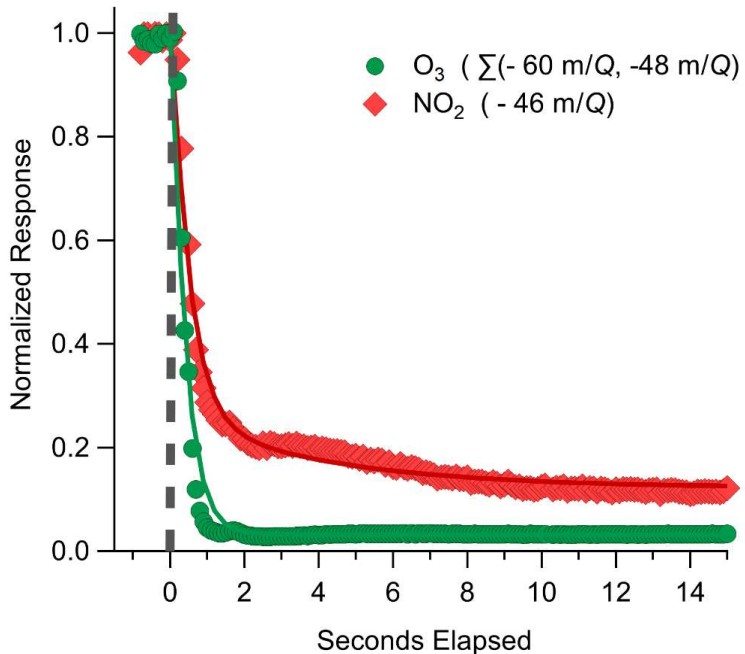

**Figure 6: Representative instrument backgrounding determination for O₃ and NO₂ where the inlet was rapidly switched from ambient sampling to an overflow with dry UHP N₂ indicated by the grey dashed line. Response is fit to a bi-exponential decay, plotted as solid lines, where the initial rapid decay ($\tau_1$) in attributed to gas evacuation of the inlet line and the second slower decay ($\tau_2$) is attributed to equilibration with the inlet walls. Best fit estimates for $\tau_1$ of O₃ and NO₂ were from 0.7 to 1.2 seconds. $\tau_2$ for O₃ was negligible at 0.3 s indicating O₃ has minimal interactions with the walls. $\tau_2$ for NO₂ was determined to be 3.2 s.**




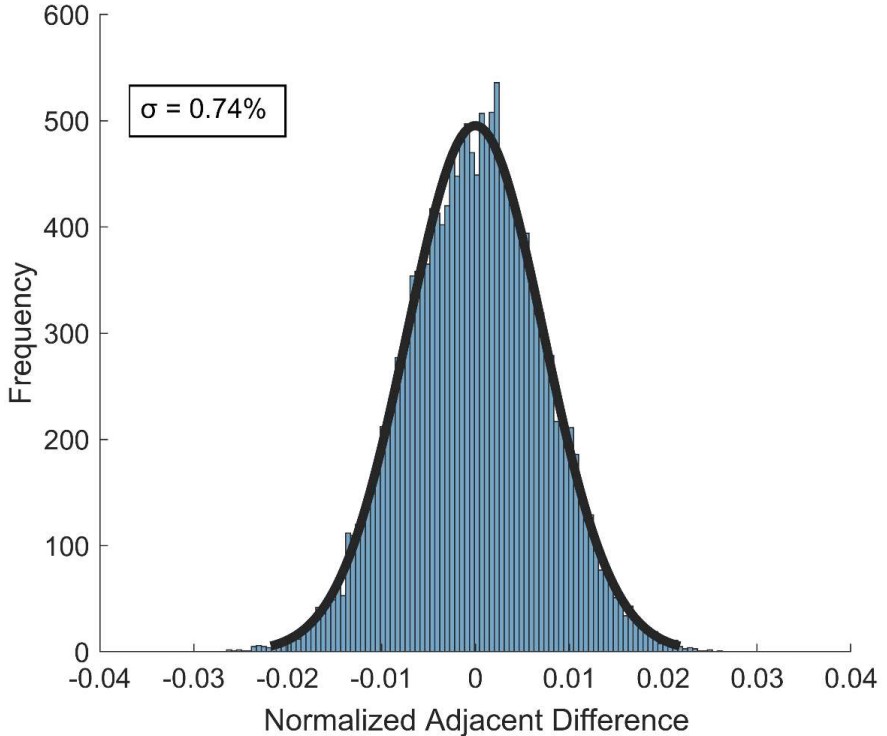

**Figure 7: Distribution of normalized adjacent differences measured at 10 Hz during a stable 27-minute ambient sampling period of**
**38 ppbv O$_3$ from Scripps Pier. The 1σ value of the distribution gives an upper limit of instrument precision of 0.74%.**



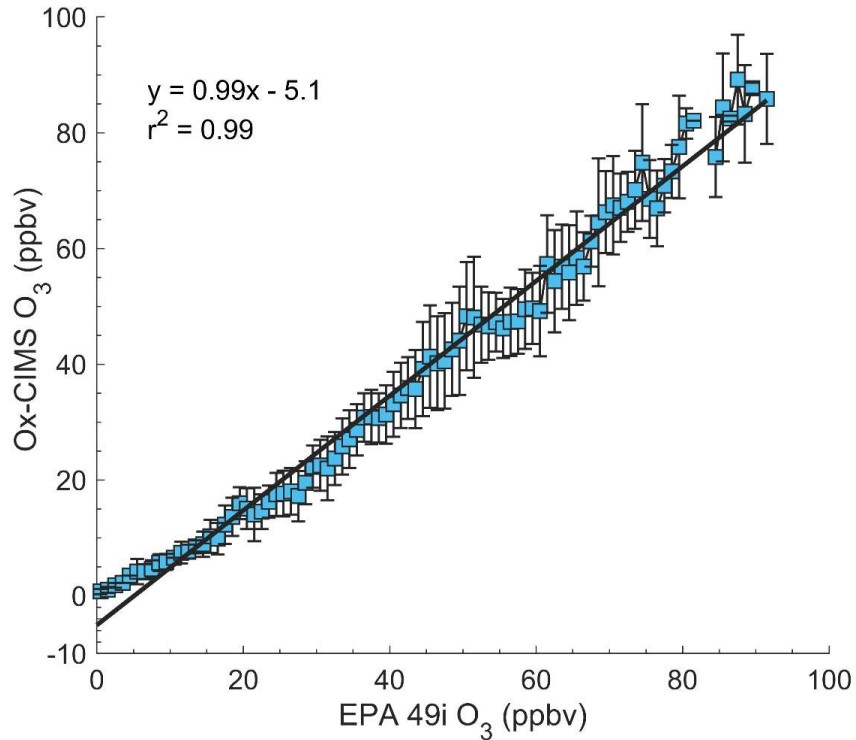


**Figure 8: Regression of 1-minute average O₃ mixing ratios from the Ox-CIMS against an EPA O₃ monitor (Thermo-Fisher 49i) binned to 1 ppbv over four weeks of ambient sampling in Zion, Illinois in May- June 2017. The solid black line is the linear least-squares regression. Error bars represent the standard deviation of each bin. Instrument agreement is strong for O₃ greater than 10 ppbv, with an apparent bias in one or both instruments below 10 ppbv.**




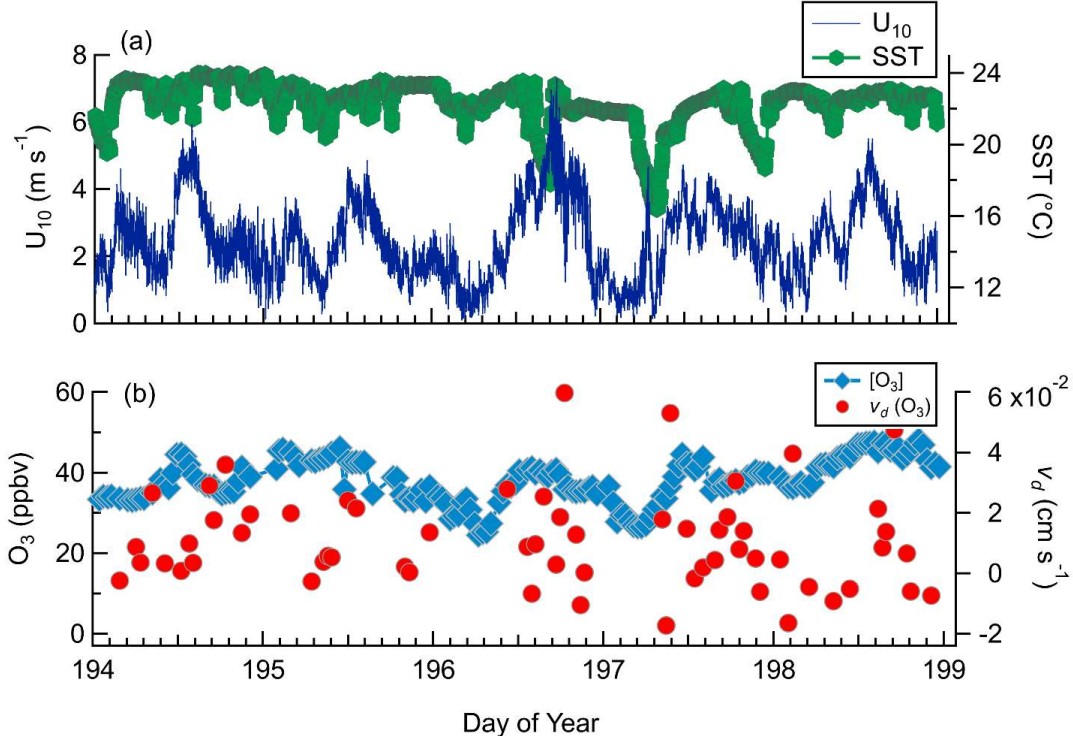

**Figure 9: Observed meteorology and O₃ mixing ratio and deposition velocities for DOY 194-199 from Scripps Pier (a) Horizontal wind speed (U₁₀) and sea-surface temperature (SST). (b) O₃ mixing ratios and vₐ(O₃).**




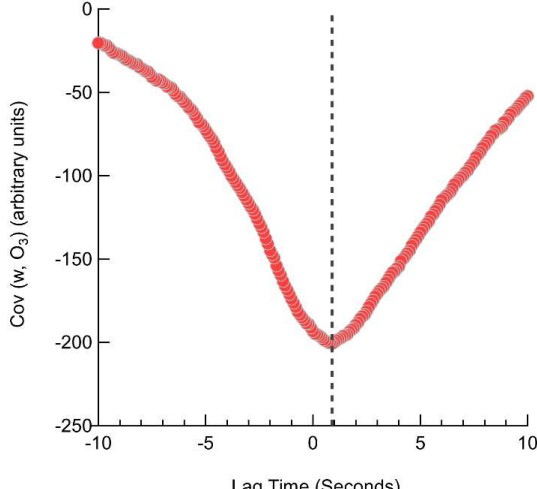

**Figure 10: Lag time determination for an individual 27-minute O₃ flux averaging period. The lag time for this flux period determined from the maximum of the covariance to be 0.9 seconds which compares reasonably with the volumetric evacuation time of the inlet of 1.7 to 2.1 seconds.**




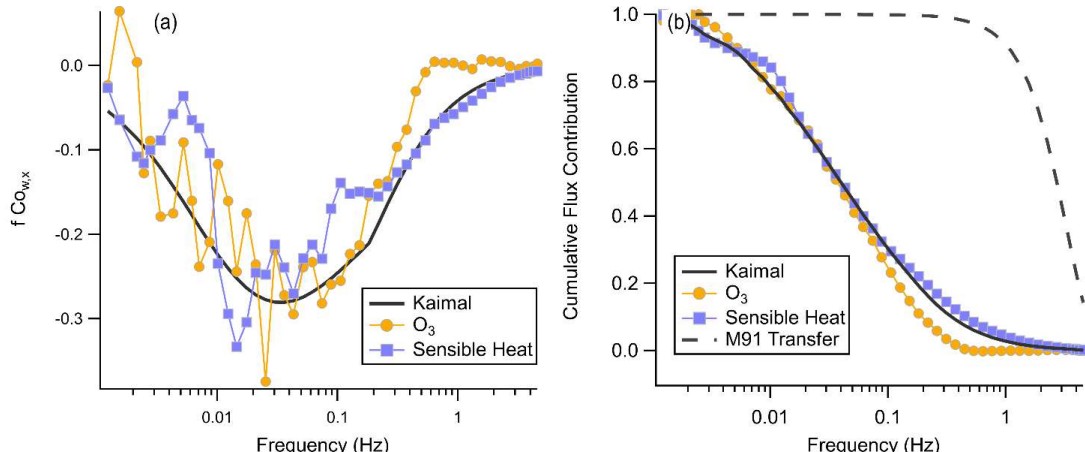

**Figure 11. (a) Mean binned frequency weighted cospectra O₃ and sensible heat flux with vertical wind from the average of two consecutive flux periods from 14:10 – 15:20 local time on July 20th. The Kaimal trace is the idealized cospectra Kaimal et al. (1972) for mean windspeed of 4.4 m s⁻¹ and an unstable atmosphere. The sensible heat trace is inverted, and the observed net sensible heat flux was positive for this period (b) Corresponding ogives for cospectra shown in (a). The M91 Transfer trace is the calculated transfer function for turbulent attenuation in a tube from (Massman, 1991).**







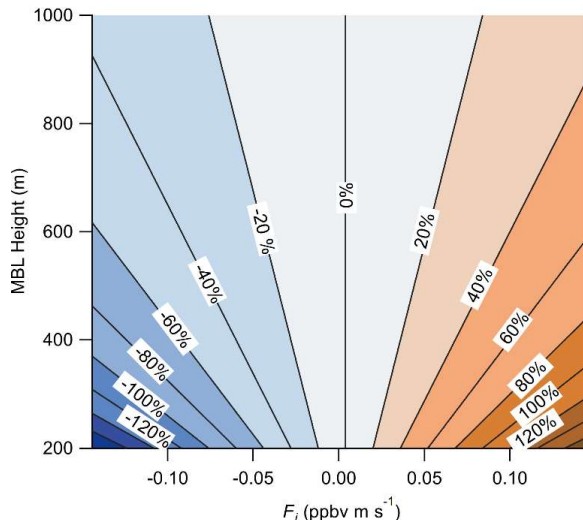

**Figure 12. Calculated percent error in the measured O₃ surface flux due to entrainment from the free troposphere as a function of the MBL height and the entrainment flux ($F_i$). Entrainment flux is the product of the free troposphere to boundary layer concentration gradient ($\Delta C$), and the entrainment velocity ($w_e$). Calculation of percent error used the Scripps Pier measuring height of 13 m, and mean surface flux of -4.4 x 10⁻³ ppbv m s⁻¹.**







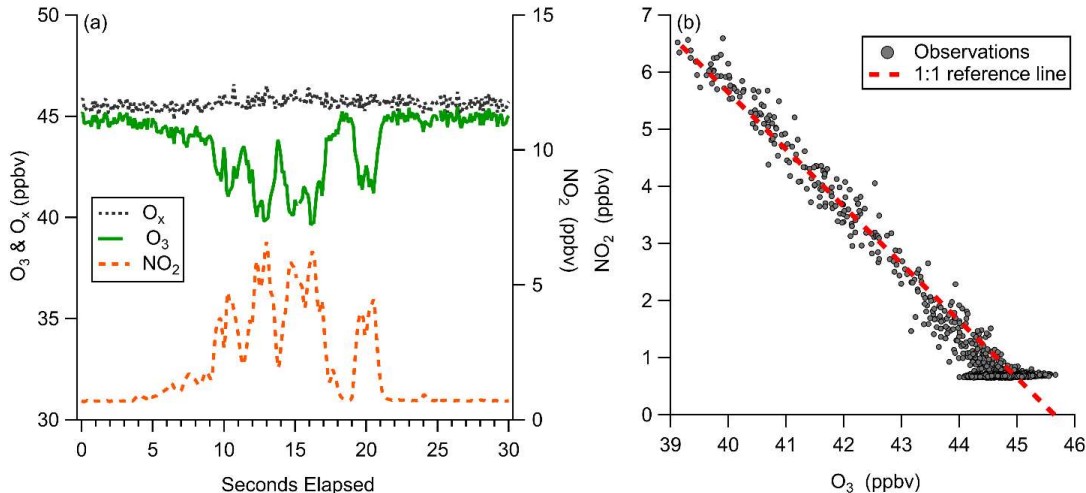

**Figure 13: Observations of ozone titration by NO emissions from a boat engine near the SIO pier. (a) 10 Hz timeseries of $O_3$, $NO_2$, and $O_x$ ($O_3 + NO_2$) demonstrating ability to capture transient titration events. (b) Regression of $O_3$ and $NO_2$ plotted with a reference line of slope -1, showing conservation of total $O_x$ at 10 Hz during a NO titration event.**

