# Peer review of "Simultaneous Detection of Ozone and Nitrogen Dioxide by Oxygen Anion Chemical Ionization Mass Spectrometry: A Fast Time Response Sensor Suitable for Eddy Covariance Measurements"

_Atmospheric Measurement Techniques, 2019_

## Referee Comment (RC1) · Mingxi Yang (Referee) · 6 Dec 2019

This paper describes the use of the oxygen anion chemical ionization mass spectrometer for simultaneous measurements of N2O and O3, with the application of eddy covariance flux measurements over the sea. It's another example of the versatile utility of the time-of-flight CIMS. The paper is generally well written and the authors have carefully considered the various aspects of data processing and interpretation. I recommend publication after they address the following mostly minor comments.

[Figure]

Abstract: Counts /s/ppt instead of ions/s/ppt

Line 53: authors introduced wet, dry, and gas-phase chemiluninescence methods here in this order, but discussed them subsequently in a different order (gas-phase, wet, dry). Suggest discussing the methods following the order of wet, dry, gas-phase

Line 230. The increase in ncps of 175% doesn't quite square up with Fig. 5 by eye. This pressure sensitivity needs to be treated with care. How precise/accurate is the pressure in the IMR controlled? On a mobile platform (e.g. ship), motion sometimes can induce a pressure fluctuation. It would be good if the instrument can keep the pressure very accurate and constant, even in the presence of motion.

Line 245. Have the authors compared this N2 background vs. simply scrubbing the measured air (e.g. with activated charcoal)? The differences in H2O and CO2 between ambient air and scrubbed air are much smaller than those between ambient air and N2. This should make the background measurements easier to interpret.

Section 2.9. authors have shown that O3 normalized sensitivity is linear (up to 80 ppbv), despite the fact that CO3- signal and O2- signal being comparable in magnitude. They have also shown that NO2 normalized sensitivity doesn't depend on O3 level. Does the O3 normalized sensitivity depend on the NO2 concentration?

Paragraph beginning on line 317. This paragraph isn't very clear. How is it that above 1e6 cps, precision no longer improves with count rate, yet "For 10 Hz averaging and count rates of 1e6 and 1e7 cps, the corresponding instrument precision is 0.75 and 2% respectively, and appears independent of count rate"?

Also, it would help the readers to spell out how the counting noise is computed.

Line 339. Without being familiar with these TOFMS or exploring the Vermeuel et al. reference, it's unclear how this O3 calibration factor is applied. You could refer readers to section 3.2.1 here.

Section 3.1 suggest adding 1-2 sentences describing how the CIMS was deployed.

Was it subsampling from a inlet manifold like on Scripps pier? Length of inlet? Instrument temperatures?

Line 342. The difference between the two instruments warrants further investigation. Even at 80 ppb the two instruments don't perfectly agree. Did the EPA monitor have a Nafion dryer to remove water vapor? What's the response time of the EPA monitor? If fairly slow, then during an O3 titration event due to NO the CIMS would initially see lower O3 than the EPA monitor at 1-minute resolution...

Line 358. The authors have not discussed how their measurements might depend on the front block and IMR temperatures. Does temperature affect the stability of the clusters in the multi-step reactions?

Also, does the use of 40 deg. nlet line have any affect on O3/heterogenous chemistry within the inlet?

Line 392. These are pretty high flow rates. I'm not familiar with the internal volumes of the mass spec, but would've expected to see a faster response time than the 0.9 s quoted here. A couple of questions: 1. Are the authors confident that the N2 overflow tube consisted of N2 only (i.e. no diffusion of ambient air into that tube)? From experience, even when using a fairly thin tube (1/8" OD) to tee into the main manifold, there can be some diffusion of ambient air into the N2 line if it's just an open tee. This can be overcome by either having the shut-off valve next to the tee, or by doing the N2 puff multiple times in succession. 2. Alternatively, could the fairly low response time be due to the multi-stage chemistry?

Eq. 4. Suggest replace KH with just H, to avoid confusion with Kt.

This sentence very confusing. Suggest rewrite: "Outliers in vd(O3) and the flux limit of detection were determined and removed for points three scaled median absolute deviations from the median. "

Line 429. 84% flux rejection is clearly not ideal. Instead of applying a simple u* threshold, I encourage the author to investigate the u* and Cd vs. wind speed relationship. This stress relationship is fairly well known over the ocean, and the authors could choose to reject O3 flux values when the measured u* or Cd is far from expected.

Line 455. The lag time determined from maximum covariance is approximately half as much as those computed from the gas evacuation. I suppose this could be due to either a time error between the O3 and wind measurement, or the fact that the inlet pressure is much lower than 1 atm (such as the volumetric flow rate is ∼2x the mass flow rate).

Line 467. One wouldn't expect the lag time to be the same as the response time. If t0= 0 represents the time when the N2 was injected into the inlet, the O3 signal should start to drop ∼1 s later, and reach 1/e of the initial value ∼1.9 s after t0.

Line 495. One way to deal with estimation of high frequency flux loss without directly using the noisy O3 cospectra would be to take an unattenuated cospectra (could be Kaimal, or could be the less noisy measured heat cospectra), attenuated it with a filter function (e.g. Eq. 7 in Bariteau et al. 2010), and compute the ratio between unattenuated and attenuated cospectra. Finally you can apply this ratio to your measured O3 flux to get the unattenuated flux. The flux loss at high frequencies is pretty obvious despite the very low wind speed. So this correction is worth characterizing well.

Line 523. This is most likely true. See www.atmos-meas-tech.net/9/5509/2016/ for example.

Section 3.7.1 it might be worth mentioning that emission of NO from other sources (e.g. ships) could also bias the O3 flux measurement. Though the authors' despiking of the O3 signal probably removed such short-term ship emission-related O3 titrations.

Section 3.7.2 My understanding is that a vertical gradient in flux does occur within the MBL when there's a large entrainment flux, but this mostly applies to the region ABOVE the 'constant flux layer' (i.e. more relevant for aircraft studies). The constant flux layer

latter is usually taken to be roughly the lowest 10% of the MBL. Within the constant flux layer, we typically assume that there isn't a vertical gradient in flux, and the measured flux = surface flux. I'm not aware of people making H2O flux measurements from a ship/buoy needing to worry about the entrainment flux, for example. Some more discussion/references on this topic would be welcomed.

Line 589. 'Within range' instead of 'in good agreement', since there's a lot of variability in previous measurements.

---

## Referee Comment (RC3) · Anonymous Referee #3 · 10 Jan 2020

General comments:

A novel method of measuring O3 and NO2 based on chemical ionization time-of-flight mass spectrometry with oxygen anion (O2-) as the reagent ion (Ox-CIMS) is developed. This new method is able to measure O3 and NO2 at fast time response and low mixing ratios, which is applicable to eddy covariance flux measurements. The authors conducted thorough characterization of the sensitivity, ion chemistry, inlet, calibration in the laboratory. They also used the instrument for the measurement of O3 vertical fluxes over the coastal ocean, via eddy covariance. Their measured flux is in good

agreement with prior studies of O3 ocean-atmosphere exchange. Potentially, fluxes for multiple species can be obtained with one measurement with the Ox-CIMS. During the same campaign, they also used a 2B ozone monitor to measure ozone, which agreed well with the Ox-CIMS measurement. The paper is well written, and I suggest publishing this work after addressing the following specific comments.

Specific comments:

Around line 138 to 153 on the discussion of CO3- ion formation, do other chemicals also form CO3-? It was mentioned early on line 119 that SO2 also forms CO3-? How to rule out that CO3- detected are not from other chemicals? Similarly, on line 215, would CO3- come from other species, rather than O3+CO2+O2-chemistry?

Line 154: Are there other interfering species that will end up as NO2- in the CIMS? Do HNO3, HONO, PAN or Organ-NO2 form NO2- with the ion chemistry? For example, on line 282, the authors mention that "A possible source of this background is from degradation of other species such as nitric acid or alkyl nitrates on the inlet walls." Did the authors do any test for interfering species?

Line 189, can the authors specify what the normalized counts are? Is it normalized to the reagent ion counts?

Section 2.8: The authors mentioned that background measurement influences the detection limit. Do they have any recommendation in improving the detection limit?

Line 572: It might be easier for readers to include the equation in the paper and cite Bariteau et al., so readers won't need to download Bariteau et al.

---

## Author Comment (AC1) · 20 Feb 2020

Response to Reviewers

Comments for all Reviewers: We thank all reviewers for their time and thoughtful comments which have substantially improved the quality and clarity of our manuscript! Below we highlight three alterations to the manuscript, based on reviewer comments, which we believe are relevant for all reviewers. These changes have not necessitated changes to any of the conclusions of this work but have meaningfully improved the quality of our data treatment.

1) Based on very helpful comments from Reviewer 1, several portions of the eddy covariance data processing and quality control have been altered. In particular, we have devised a friction velocity filter based on agreement between our measured values and calculated values using the NOAA COARE v3.6 bulk flux algorithms, rather than a fixed friction velocity threshold. This has increased the number of valid quality controlled flux periods and improved our ensemble flux LOD. We have also applied a frequency attenuation correction to our flux values which resulted in a mean increase in $v_d(O_3)$ of 4.5%. These changes necessitated updating all reported EC flux values in the text and in Table 2, and Figures 6, 9, 12, S9, and S12. The updated mean campaign $v_d(O_3)$ is 0.013 (changed from 0.011 cm s$^{-1}$ in the original manuscript) and the ensemble LOD is 0.0027 cm s$^{-1}$ (changed from 0.0042 cm s$^{-1}$). The total number of valid quality-controlled flux observations is now 246 (changed from 151). Further details on each of these corrections are described in specific responses to Reviewer 1. We believe these changes, made directly following recommendations of the reviewer, have significantly enhanced the quality of our manuscript.

2) Also based on helpful comments from Reviewer 1 who noted that our determined response time of 0.9 s seemed longer than expected, we have revisited our determination of the instrument response throughout the campaign. It became apparent that infrequently the automated bi-exponential decay algorithm we applied over weighted the second exponential and gave poor fitting results. Instrument response times for $O_3$ have been recalculated using a single exponential decay fit which provides a better fit to the data. Calculated fits were manually verified for the full campaign. The updated response time is faster than the original reported value (0.28 s (0.25 to 0.31 s 95% confidence bounds), previously 0.9 s). Review of this response time calculation was also motivated by its direct use in the newly implemented frequency attenuation calculation described in comment number 1. This updated response time value was also used to calculate an updated value of the cutoff frequency (now 0.57 Hz, previously 0.18 Hz). This response time is more in keeping with expectations given that $O_3$ is a "non-sticky" gas and the high flow rates and low volumes in our instrument. The use of a single-exponential decay also follows the method of Bariteau et al. (2010) for the characterization of their fast-response chemiluminescence $O_3$ sensor. These values have been updated throughout the text. Additional discussion is provided in responses to Reviewer 1.

3) Based on private communications during the review process, we have included additional caveats in Section 5 regarding the conservation of total $O_x$ observed during an $O_3$ titration from local NO emissions. In particular, it was noted that engine $NO_x$ emissions cannot always be assumed to be exclusively as NO. Ship emissions have shown NOx to NO2

emission ratios on the order of 10%. If some emissions are indeed in the form of $NO_2$, the reported conservation of total $O_x$ in section 5 would be partially spurious resulting from various compensating errors. We have added some discussion of this important point.

The manuscript has been revised to state in Section 4 : "This analysis assumes that there were no direct $NO_2$ emissions during the titration event. A $NO_2$ to $NO_x$ emission ratio of 0.08 was observed for ship emissions from diesel motors on inland shipping vessels (Kurtenbach et al., 2016). Without additional knowledge amount the $NO_x$ emission source during this event, the observed conservation of total $O_x$ could be partially driven by compensating errors within 10%."

Responses to individual reviewer comments follow. Reviewer comments are reproduced in italic black font. Author responses are shown in regular blue font. Text added to the manuscript is underlined and text removed from the manuscript has a strikethrough.

Response to Reviewer 1:

*This paper describes the use of the oxygen anion chemical ionization mass spectrometer for simultaneous measurements of N2O and O3, with the application of eddy covariance flux measurements over the sea. It's another example of the versatile utility of the time-of-flight CIMS. The paper is generally well written and the authors have carefully considered the various aspects of data processing and interpretation. I recommend publication after they address the following mostly minor comments.*

We are very grateful to the reviewer for their thorough review and numerous suggestions which have improved the quality of our data treatment and our manuscript as a whole. Below we address specific reviewer comments.

*Abstract: Counts /s/ppt instead of ions/s/ppt*
All instances of ions $s^{-1}$ $pptv^{-1}$ have been changed to counts $s^{-1}$ $pptv^{-1}$ or cps $pptv^{-1}$ (or ncps $pptv^{-1}$, etc.) as appropriate in the abstract and throughout the text to maintain consistency. We had used these terms interchangeably which was not made clear.

*Line 53: authors introduced wet, dry, and gas-phase chemiluninescence methods here in this order, but discussed them subsequently in a different order (gas-phase, wet, dry). Suggest discussing the methods following the order of wet, dry, gas-phase*

This point is well taken and the text has been rearranged as suggested but the content is unchanged.

*Line 230. The increase in ncps of 175% doesn't quite square up with Fig. 5 by eye. This pressure sensitivity needs to be treated with care. How precise/accurate is the pressure in the IMR controlled? On a mobile platform (e.g. ship), motion sometimes can induce a pressure fluctuation. It would be good if the instrument can keep the pressure very accurate and constant, even in the presence of motion.*

We thank the reviewer for catching this arithmetic error. The increase in absolute numbers was from 4.37E5 to 6.97E5 ncps. This corresponds to a 60% increase, or a signal at 95 mbar that is ~160% of the signal at 70 mbar. This value has been corrected and the text slightly clarified.

So far we have focused on ground based deployments at fixed sampling sites for the Ox-CIMS and so have not made significant efforts to ensure highly accurate pressures. The standard deviation of IMR pressure during individual flux sampling periods were typically from 0.1-0.3 mbar which would have a negligible impact on our observations. Methods for accurate and constant pressure control for CIMS instruments have been developed for airborne studies (i.e Lee et al. (2014)) and should be directly translatable to this instrument. This type of pressure control would be a useful addition before any airborne or ship-based deployments of the Ox-CIMS.

The manuscript has been revised to state: The normalized signal of $O_3$ increases by  60% at an IMR pressure of 95 mbar compared to 70 mbar  when sampling a constant $O_3$ source of 35 ppbv.

*Line 245. Have the authors compared this N2 background vs. simply scrubbing the measured air (e.g. with activated charcoal)? The differences in H2O and CO2 between ambient air and scrubbed air are much smaller than those between ambient air and N2. This should make the background measurements easier to interpret.*

The used of a scrubber will absolutely be explored before any future deployments. We also intend to explore the development of catalytic zero air generator to overflow the inlet which would maintain near ambient CO2 and H2O.  In this initial deployment we had also aimed to measure a variety of other trace gases detectable with the Ox-CIMS not discussed in this manuscript. It was decided that the N2 background was the most versatile method for zeroing all species of interest even if it necessitated additional considerations for $O_3$. Future deployments could implement a combination of scrubber-based zeros for $O_3$ and N2 overflow zeros for other species to remove this issue.

*Section 2.9. authors have shown that O3 normalized sensitivity is linear (up to 80 ppbv), despite the fact that CO3- signal and O2- signal being comparable in magnitude. They have also shown that NO2 normalized sensitivity doesn't depend on O3 level. Does the O3 normalized sensitivity depend on the NO2 concentration?*

We have not directly assessed the dependence of O3 normalized sensitivity on NO2 but this will be a valuable future laboratory experiment, especially if we intend to sample in an urban high $NO_x$ environment. We can note that the ambient normalized $O_3$ signal was insensitive to the high concentration formic acid standard additions described in this deployment which we expect to behave similarly to NO2.

*Paragraph beginning on line 317. This paragraph isn't very clear. How is it that above 1e6 cps, precision no longer improves with count rate, yet "For 10 Hz averaging and count rates of 1e6 and 1e7 cps, the corresponding instrument precision is 0.75 and 2% respectively, and appears independent of count rate"? Also, it would help the readers to spell out how the counting noise is computed.*

We agree this section was confusing as written and have improve phrasing to clarify this discussion. The quoted sentence was primarily intended to show that from 1E6 to 1E7 cps instrument precision did not continue to improve with count rate. Instrument precision below 1E6 cps did improve with count rate as expected when precision is controlled by counting noise. The meaning of the observed precision at 1E6 and 1E7 cps being specifically 0.75 and 2% is not clear and is again only meant to highlight that precision is no longer improving with count rate as expected for counting noise. The equation for counting noise has been added which is $\frac{\sqrt{N}}{N}$ where N is the number of counts during the observation period.

The manuscript has been revised to state: "From this assessment, precision was observed to improve approximately linearly in a log-log scaling for count rates between 1 x $10^3$ and 1 x $10^6$ cps (Fig. S8) as expected in the case where counting noise drives instrument precision. Above 1

x $10^6$ cps there is an apparent asymptote where precision no longer improves with count rate.  The counting noise limited 10 Hz precision for $10^6$ and $10^7$ cps  are 0.32% and 0.1% respectively, while the measured values were 0.75 and 2%. The counting noise limited precision is calculated as $\sqrt{N}/N$ where $N$ is the number of counts during the integration time."

*Line 339. Without being familiar with these TOFMS or exploring the Vermeuel et al. reference, it's unclear how this O3 calibration factor is applied. You could refer readers to section 3.2.1 here.*
*and*
*Section 3.1 suggest adding 1-2 sentences describing how the CIMS was deployed. Was it subsampling from a inlet manifold like on Scripps pier? Length of inlet? Instrument temperatures?*

Based on discussion from Reviewers 1 and 2, significant details have been added back to this section rather than requiring the reader to refer to Vermeuel et al. (2019).

The manuscript has been revised to state: Section 3.1: "The Ox-CIMS was located on the roof of a trailer (approx. 5 m above ground) and sampled through a 0.7 m long, 0.925 cm i.d., PFA inlet. The inlet was pumped at flow rate of 18-20 slpm from which the Ox-CIMS subsampled at 1.5 slpm. Temperature and RH were recorded inline downstream of the subsampling point. The Ox-CIMS sampling point was approximately 10 m horizontally from the Thermo-Fisher 49i and sampled at approximately equal heights.  Instrument backgrounds of the Ox-CIMS were determined every 70 minutes by overflowing the inlet with dry UHP $N_2$. Calibration factors  were determined by  the in-field continuous addition of a C-13 isotopically labelled formic acid standard to the tip of the inlet. Laboratory calibrations of the Ox-CIMS to formic acid and $O_3$ as a function of specific humidity were determined immediately pre- and post-campaign and were used to calculate a humidity dependent sensitivity of $O_3$ relative to formic acid. That relative sensitivity was then used to determine the in-field sensitivity to $O_3$ by scaling field sensitivities of formic acid from the continuous additions. Full details of this deployment and calibration methods are described in Vermeuel et al., (2019)."

*Line 342. The difference between the two instruments warrants further investigation. Even at 80 ppb the two instruments don't perfectly agree. Did the EPA monitor have a Nafion dryer to remove water vapor? What's the response time of the EPA monitor? If fairly slow, then during an O3 titration event due to NO the CIMS would initially see lower O3 than the EPA monitor at 1-minute resolution.*

We agree that there remain open questions in this instrument comparison which are difficult to assess with the data available. While agreement between the two instruments is not perfect, we believe they show strong agreement for field sampling in a complex environment, especially as we are comparing to a monitoring grade instrument. The Thermo Fischer 49i (EPA O3 monitor) was not equipped with a Nafion drier and the manufacturer quoted response time for the instrument alone is 20 seconds. An additional unquantified response time of similar magnitude was likely present due to the instrument sampling from a long, wide diameter inlet at a low flow rate. Unfortunately, specific details of EPA O3 monitor inlet configuration were not recorded

during this deployment, as it was not originally devised as an $O_3$ intercomparison study. Taken together, the EPA O3 monitor was likely subject to interferences from water vapor and other species and had a slower response to titration events and changes in air masses. We also note that $O_3$ mixing ratios exceeding 80 ppbv were only sampled on three afternoons during the study, which were driven by highly polluted urban airmasses from Chicago, making robust comparison between the instruments at high mixing ratios a challenge. An instrument intercomparison study with a research grade $O_3$ instrument, such as an NO chemiluminescence sensor, rather than a monitoring grade instrument like the Thermo Fischer 49i will be valuable in further characterizing the Ox-CIMS.

*Line 358. The authors have not discussed how their measurements might depend on the front block and IMR temperatures. Does temperature affect the stability of the clusters in the multi-step reactions? Also, does the use of 40 deg. nlet line have any affect on O3/heterogenous chemistry within the inlet?*
The inlet temperature of 40°C was primarily selected to ensure that the inlet line was always held above ambient temperatures to prevent any potential condensation of water vapor in the line and to ensure consistent sampling conditions. Additional impacts on heterogeneous chemistry or ion-adduct stability were not quantitatively assessed. We speculate that these effects would be minor due to being only slightly above ambient temperatures (ca 25°C).

*Line 392. These are pretty high flow rates. I'm not familiar with the internal volumes of the mass spec, but would've expected to see a faster response time than the 0.9 s quoted here. A couple of questions: 1. Are the authors confident that the N2 overflow tube consisted of N2 only (i.e. no diffusion of ambient air into that tube)? From experience, even when using a fairly thin tube (1/8" OD) to tee into the main manifold, there can be some diffusion of ambient air into the N2 line if it's just an open tee. This can be overcome by either having the shut-off valve next to the tee, or by doing the N2 puff multiple times in succession. 2. Alternatively, could the fairly low response time be due to the multi-stage chemistry?*

Based on the reviewer's comments we have revisited our determination of the instrument response throughout the campaign. It became apparent that the automated bi-exponential decay algorithm we applied frequently over weighted the second exponential and gave poor fitting results. Instrument response times for $O_3$ have been recalculated using a single exponential decay fit which provides a better fit to the data. Calculated fits were manually verified for the full campaign. The updated response time is faster than the original reported value (0.28 s (0.25 to 0.31 s 95% confidence bounds), previously 0.9 s). Review of this response time calculation was also motivated by its direct use in the newly implemented frequency attenuation calculation described in comment number 1. This updated response time value was also used to calculate an updated value of the cutoff frequency (now 0.57 Hz, previously 0.18 Hz). This response time is more in keeping with expectations given that $O_3$ is a "non-sticky" gas and the high flow rates and low volumes in our instrument. The use of a single-exponential decay also follows the method of Bariteau et al. (2010) for the characterization of their fast-response chemiluminescence $O_3$ sensor. These values have been updated throughout the text. See below for a plot of single-exponential decays fits determined throughout the full campaign shown as grey traces, and the binned mean decay curve shown in the red trace. The dashed horizontal line is at 1/e which corresponds to the response time.

[Figure]

*Eq. 4. Suggest replace KH with just H, to avoid confusion with Kt.*
This terminology has been changed as suggested.

*Line 427. This sentence very confusing. Suggest rewrite: "Outliers in vd(O3) and the flux limit of detection were determined and removed for points three scaled median absolute deviations from the median. "*

We appreciate this note and have updated the text as suggested.

*Line 429. 84% flux rejection is clearly not ideal. Instead of applying a simple u* thresh- old, I encourage the author to investigate the u* and Cd vs. wind speed relationship. This stress relationship is fairly well known over the ocean, and the authors could choose to reject O3 flux values when the measured u* or Cd is far from expected.*
We are grateful to the reviewer for this suggestion which we have implemented as follows. We have calculated the expected $U_*$ for each flux period using the NOAA COARE v3.6 bulk flux algorithms, using observed wind speeds, SST, air temperature, and humidity as inputs. Flux periods were rejected if the calculated $U_*$ was within 50% of the observed value. This filter resulted in a total of N = 246 valid flux points, mean $v_d(O_3)$ =0.013 cm s$^{-1}$ and ensemble LOD of 0.0027 cm s$^{-1}$. We note that we are relatively insensitive to the specific threshold applied, selecting a threshold of 40% agreement yields N=191, $v_d(O_3)$ =0.0137 cm s$^{-1}$, and LOD =0.0032 cm s$^{-1}$. A threshold of 30% agreement yields N=138, $v_d(O_3)$ =0.0145 cm s$^{-1}$, and LOD =0.0027 cm s$^{-1}$. We have selected the threshold that minimizes the LOD while maintaining more valid flux points than the fixed $U_*$ filter of 0.1 cm s$^{-1}$. We would also like to point out that the 84% flux rejection included the wind direction filter which may not have been clear. Over 30% of flux points were immediately rejected based on winds not coming from the ocean. We also note that even in the case of the 30% agreement threshold the ensemble LOD was improved compared to our original fixed $U_*$ threshold. This suggests that this relative threshold does a better job of filtering low quality flux periods.

The text has been updated substantially in Section 3.3 to account for these changes.

*Line 455. The lag time determined from maximum covariance is approximately half as much as those computed from the gas evacuation. I suppose this could be due to either a time error between the O3 and wind measurement, or the fact that the inlet pressure is much lower than 1 atm (such as the volumetric flow rate is _2x the mass flow rate).*

The slight disagreement of these values was also notable to us. We believe the source of this is likely due to a timing error as suggested by the reviewer as the software saving the mass spectrometer and anemometer data signals were not optimized to have highly precise timing. The inlet pressure is also a likely contributing factor as suggested. A quick calculation of expected pressure drop in our inlet suggest that pressure at the Ox-CIMS subsampling point is ~150 mbar lower than ambient. As noted by the reviewer, this pressure drop will increase the volumetric at a given measured mass flow rate. Taken together we expect these factors to explain our observed lag time. We note that applying a fixed lag time corresponding to the volumetric residence time results in a calculated $v_d(O_3)$ of 0.012 cm s$^{-1}$ compared the our value of 0.013 cm s$^{-1}$ using the fixed lag time.

*Line 467. One wouldn't expect the lag time to be the same as the response time. If t0= 0 represents the time when the N2 was injected into the inlet, the O3 signal should start to drop _1 s later, and reach 1/e of the initial value _1.9 s after t0.*

We agree that the total instrument response time should be the sum of the volumetric clearing time of the inlet and the instrument response time. In our case the determined lag time for eddy covariance also appears to be influenced by software timing consideration as noted above. In our determination of the instrument response time, we treated $t_0$ as the point immediately before the signal responded to the $N_2$ overflow. This is done to separate out determinations of the inlet lag time and the instrument response time. This follows the method used by Bariteau et al (2010), which based on our determination on. Additionally, the position of the solenoid valve controlling the $N_2$ overflow was only recorded at 1Hz which makes it difficult to resolve the lag time based on the when the solenoid was flipped. This is also why we were no able to determine the instrument lag time using the "puff method" described in Bariteau et al. Calculating the response time based on when the signal begins to respond rather when the solenoid flipped removes that imprecision.

Also, as noted in the comment we made above we have since revisited our determination of the instrument response time, with a new mean value of 0.28 s based on a single-exponential decay fit.

*Line 495. One way to deal with estimation of high frequency flux loss without directly using the noisy O3 cospectra would be to take an unattenuated cospectra (could be Kaimal, or could be the less noisy measured heat cospectra), attenuated it with a filter function (e.g. Eq. 7 in Bariteau et al. 2010), and compute the ratio between unattenuated and attenuated cospectra. Finally you can apply this ratio to your measured O3 flux to get the unattenuated flux. The flux loss at high frequencies is pretty obvious despite the very low wind speed. So this correction is worth characterizing well.*

We thank the reviewer for the push to treat this attenuation correction properly. We have implemented a frequency correction following this suggestion. Briefly, we calculated the unattenuated Kaimal cospectra (assuming an unstable boundary layer) for each individual flux measurement period. We then applied the low pass filter function described by Bariteau et al. (2010) using a fixed response time of 0.28 s to determine the attenuated cospectra. The ratio of the attenuated to unattenuated cospectra was then taken and used as a correction factor. The net effect of this correction was an increase of $v_d(O_3)$ of 4% (increased from 0.0127 to 0.0132 cm s$^{-1}$). The manuscript has been changed in many locations to account for this change.

The manuscript has been revised significantly in Section 3.4 to reflect these changes.

*Line 523. This is most likely true. See www.atmos-meas-tech.net/9/5509/2016/ for example.*

We thank the reviewer for bringing this to our attention. We have added a short discussion and citation to the text.

The manuscript has been revised to state: Line 523: " This has been demonstrated in an EC study utilizing a closed path $H_2O$ sensor for EC flux measurements (through an 18 m long, 0.635 cm i.d. inlet, pumped at 18 slpm, comparable to the inlet used in this study) which showed complete attenuation above 0.1 Hz and overall attenuation of ~80% of the $H_2O$ (latent heat) flux (Yang et al., 2016). However, without a direct measure of water vapor fluctuations collocated with the Ox-CIMS this is difficult to  definitively rule out in our measurement."

*Section 3.7.1 it might be worth mentioning that emission of NO from other sources (e.g. ships) could also bias the O3 flux measurement. Though the authors' despiking of the O3 signal probably removed such short-term ship emission-related O3 titrations.*
We agree that this point should be stated explicitly. As stated by the reviewer we expect the combination of the despiking and stationarity criteria will do a suitable job of filtering out the very short and longer titration events respectively.

The manuscript has been revised to state: Section 3.7.1: "There is also potential for short term anthropogenic emissions of NO (such as from a boat engine passing by the sensor) to create a flux divergence term. We expect that the combination of signal despiking and the flux stationarity criteria described in Section 3.3 will minimize the impact of this potential divergence term. Despiking will remove most short term (<1 s) emission events and the stationarity criteria will filter out any period where longer term titration events cause large changes in the observed flux within a flux measurement period."

*Section 3.7.2 My understanding is that a vertical gradient in flux does occur within the MBL when there's a large entrainment flux, but this mostly applies to the region ABOVE the 'constant flux layer' (i.e. more relevant for aircraft studies). The constant flux layer latter is usually taken to be roughly the lowest 10% of the MBL. Within the constant flux layer, we typically assume that there isn't a vertical gradient in flux, and the measured flux = surface flux. I'm not aware of people making H2O flux measurements from a ship/buoy needing to worry about the entrainment flux, for example. Some more discussion/references on this topic would be welcomed.*

Our exploration of flux divergence was inspired by the aircraft observations of Lenschow et al., (1982) which showed a linear flux divergence with altitude for measurements at 15, 60, and 325 m over the ocean. The boundary layer height during those flights was *ca* 1.2 km. The flux observations at the 15 and 60 m measurement heights showed strong divergence despite being in within the constant flux layer (based on the lowest 10% rule). Lenschow et al. (1982), argue that because the surface flux of $O_3$ is much smaller than the entrainment flux ($w_e = 0.8$ cm s$^{-1}$), it is acutely sensitive to boundary layer processes. Faloona et al., (2005) also showed a linear flux divergence in $O_3$ down to 100m with a boundary layer height of ~800 m. There it is unclear if the lowest level leg was within the constant flux layer but still suggests that entrainment driven flux divergence should be considered. We realize that not citing and discussing Lenschow et al. (1982) in this section was an oversite and have revised the text to include discussion of their observations.

We also made our calculations directly following the equations in Blomquist et al. (2010) who assessed impacts of entrainment on near surface ($z_i$ =18 m) observations of DMS flux. Blomquist et al. (2010) applied a linear extrapolation from the boundary layer height to the surface and did not invoke a constant flux layer. Based on these results we believe there is value in exploring this potential source of bias in our observations even if we do not have direct constraints on its magnitude.

The manuscript has been revised to add: "Lenschow et al., (1982) presented aircraft observations of $O_3$ deposition over the Gulf of Mexico at heights of 15, 60, and 325 m which showed a strong flux gradient term driven by entrainment from the free troposphere. The boundary layer height ($z_i$) during those flights was approximately 1.2 km, suggests a strong flux gradient was present even within the surface layer (approximated as the lowest 10% of the boundary layer)."

We also note the correction of a typo on line 571. $z_i$ is the boundary layer height and $z$ is the measurement height. These definitions were reversed in the text but were used correctly in Eq. 9 and all calculations following it.

*Line 589. 'Within range' instead of 'in good agreement', since there's a lot of variability in previous measurements.*
This point is well taken, and the text has been updated using the reviewer's phrasing

References:

Bariteau, L., Helmig, D., Fairall, C. W., Hare, J. E., Hueber, J. and Lang, E. K.: Determination of oceanic ozone deposition by ship-borne eddy covariance flux measurements, Atmos. Meas. Tech., 3(2), 441–455, doi:10.5194/amt-3-441-2010, 2010.

Blomquist, B. W., Huebert, B. J., Fairall, C. W. and Faloona, I. C.: Determining the sea-air flux of dimethylsulfide by eddy correlation using mass spectrometry, Atmos. Meas. Tech., 3(1), 1–20, doi:10.5194/amt-3-1-2010, 2010.

Faloona, I., Lenschow, D. H., Campos, T., Stevens, B., van Zanten, M., Blomquist, B., Thornton, D., Bandy, A. and Gerber, H.: Observations of Entrainment in Eastern Pacific Marine Stratocumulus Using Three Conserved Scalars, J. Atmos. Sci., 62(9), 3268–3285, doi:10.1175/JAS3541.1, 2005.

Lee, B. H., Lopez-Hilfiker, F. D., Mohr, C., Kurtén, T., Worsnop, D. R. and Thornton, J. A.: An iodide-adduct high-resolution time-of-flight chemical-ionization mass spectrometer: Application to atmospheric inorganic and organic compounds, Environ. Sci. Technol., 48(11), 6309–6317, doi:10.1021/es500362a, 2014.

Lenschow, D. H., Pearson, R. and Stankov, B. B.: Estimating the ozone budget in the boundary layer by use of aircraft measurements of ozone eddy flux and mean concentration, J. Geophys. Res., 86(C8), 7291, doi:10.1029/JC086iC08p07291, 1981.

Responses to Reviewer 2:

*This paper describes the development and application of an oxygen anion chemical ionization mass spectrometry approach for directly measuring the flux of ozone and nitrogen dioxide. Of particular note, is the successful application of this technique in the marine boundary layer where the magnitude of O3 and NO2 fluxes is low. The authors describe thoughtful and extensive laboratory characterization, comparison with traditional measurements in the field, initial deployment for flux measurements, and*
*data analysis and correction. The paper is well organized and clearly written. The comments and suggestions below are meant to improve an already very good paper.*

We thank the reviewer for their supportive and thoughtful comments! Replies to all specific reviewer comments are below.

**Specific Comments:**
*L35-39: Is there any experimental evidence that vd-O3 depends on factors beyond wind speed and SST? e.g., surface ocean composition? Could such factors contribute to the order of magnitude range noted in L35? Do any measurements exist over snow ice? Or freshwater versus seawater?*

Ocean surface composition is known to be a controlling factor in $v_d(O_3)$, with Iodide and various dissolved organic compounds (DOCs) being the primary contributors. Parameterizations to SST are primarily made as a proxy for iodide which is globally directly correlated with SST. In nearly all field measurements of $v_d(O_3)$, DOC and iodide were not measured. Instead laboratory studies of O$_3$ deposition to water containing DOC and iodide are used to model their role in ambient O$_3$ deposition. To our knowledge there is one EC measurement of O$_3$ deposition to freshwater (Weseley et al. (1980)) who reported a $v_d(O_3)$ of 0.01 cm s$^{-1}$. Incidentally we have a manuscript in progress applying the Ox-CIMS reported here for deposition measurements to lake water. Observations to snow vary widely but current best estimates suggest that deposition is slow (0-0.01 cm s$^{-1}$). Mention of these values has been added to the text. We have also added some discussion to the role of iodide and DOC in controlling O$_3$ to the text here.

The manuscript has been revised to state: Starting at Line 35: "There is only one reported study of O$_3$ deposition to freshwater, which showed $v_d(O_3)$ of 0.01 cm s$^{-1}$ (Wesely et al., 1981).. Measured deposition rates to snow and ice vary widely, with most observations of $v_d(O_3)$ from 0 to 0.2 cm s$^{-1}$, while models suggest $v_d(O_3)$ from 0 to 0.01 cm s$^{-1}$ (Helmig et al., 2007). Reactions of O$_3$ with iodide and dissolved organic compounds (DOC) in the ocean are known to play a controlling role in setting $v_d(O_3)$ and may explain some of the variability in observations (Chang et al., 2004; Ganzeveld et al., 2009). However, these quantities have not typically been measured during field studies of $v_d(O_3)$."

*L53: Is it worth pointing out explicitly here, for the non-expert reader, that UV photometric detection of O3 doesn't provide fast enough time response for flux measurements?*
We agree this is important to state explicitly and have added this discussion as suggested.

The manuscript has been revised to state: Starting at Line 53: "Due to this constraint, standard O$_3$ monitoring instruments which utilize UV-absorption detection do not have suitable time response or precision for EC measurements and ozone flux measurements have primarily utilized fast response chemiluminescence sensors."

*L110-111: Which type of TofWerk ToF is used here? e.g., HToF or CToF?*
We use a CToF with resolution of *ca.* 950 at the CO3- product (-m/Q 60). We have updated the text to state this more explicitly.

The manuscript has been revised to state: Line 110: "the ion beam for entry into the compact ToF mass analyzer (CToF,…)"

*L126-127: Any reference for "a wide class or molecules"? Or, can the authors be slightly more specific? e.g., hydrocarbons/oxygenates/S-containing/N-containing etc*

The text as written was likely too vague and we have removed the phrase "a wide class of molecules" and instead point the reader to a review paper which compiled electron affinity measurements and calculations. This reference provides the reader with a starting place to easily asses if the Ox-CIMS has potential to measure a molecule of interest.

The manuscript has been revised to state: Line 126: "…resulting in a relatively non-specific reagent ion chemistry  (see Rienstra-Kiracofe et al., (2002) for a compilation of molecular E.A. values)."

*L181-189 (and elsewhere): Is ions/s used equivalently with counts/s? Does one quantity rely on the calibration of the single ion signal? A clear definition would be helpful.*

Thank you for pointing out this ambiguity. Ions s$^{-1}$ and counts s$^{-1}$ are equivalent and were used interchangeably throughout the text. We have changed all instances of ion s$^{-1}$ or related expressions throughout the text to counts s$^{-1}$ (or cps/ncps as appropriate) for clarity.

*L187: Does this mean that signals are normalized to a fixed value? Or normalized to the variable signal for a reagent ion? Or, does this refer to the signal/pptv you would obtain for a reagent ion signal of 1e6 ions/s? Clarification would be helpful.*

We normalize signals by scaling to a fixed total reagent ion signal of 1E6 cps. This follows the standard approach in many CIMS applications. Based on suggestions from Reviewers 2 and 3, this section has been clarified.

The manuscript has been revised to state: Line 187: "Sensitivity values can be normalized by scaling all signals to a fixed total reagent ion signal  of $1 \times 10^6$ cps  to isolate the sensitivity component controlled by reagent ion chemistry, separate from  changes in instrument performance due to decay in the ion source or other factors. The total reagent ion signal is taken as the sum of the $O_2^-$ and $O_2(H_2O)^-$ signals."

*L223-224: Reference or supplemental data for this statement?*

This statement as written was likely an overstatement based on the amount of evidence we currently have. Laboratory calibrations of formic acid, $H_2O_2$, $NO_2$, and nitric acid have shown insignificant difference when performed in $N_2$ or in ambient air. Beyond that we have not performed a robust set of experiments to specifically rule out the involvement of CO2 in the detection of other analytes. We have revised the text to be less definitive and acknowledge that only a small set of molecules have been specifically calibrated.

The manuscript has been revised to state: Line 223: "No other analytes that we have calibrated for  with the Ox-CIMS (HCOOH, $HNO_3$, $H_2O_2$) have shown a $CO_2$ mixing ratio dependence,  suggesting that $CO_2$ may be  uniquely involved in the detection of $O_3$  and is not a general feature of the oxygen-anion chemistry."

*L292-293: Are signals only normalized to the reagent ion when the reagent ion is lower than the analyte signal?*

Ambient data was normalized for all analyte signal magnitudes. This is necessary for applying laboratory humidity dependent sensitivities to ambient observations despite slow drift in instrument performance.

*L331 (section 3.1): The next section (3.2) contains quite a lot of detail on instrument set up in the field (temperatures, inlet etc.), but relatively little information is given here. More detail would be useful.*

And

*L339-341: Somewhat more detail on this scaling and how it is assessed/applied is warranted here, rather than relying heavily on the Vermeuel reference.*

Based on discussion from Reviewers 1 and 2, significant details have been added back to this section rather than requiring the reader to refer to Vermeuel et al. (2019).

The manuscript has been revised to state: Section 3.1: "The Ox-CIMS was located on the roof of a trailer (approx. 5 m above ground) and sampled through a 0.7 m long, 0.925 cm i.d., PFA inlet. The inlet was pumped at flow rate of 18-20 slpm from which the Ox-CIMS subsampled at 1.5 slpm. Temperature and RH were recorded inline downstream of the subsampling point. The Ox-CIMS sampling point was approximately 10 m horizontally from the Thermo-Fisher 49i and sampled at approximately equal heights. Instrument backgrounds of the Ox-CIMS were determined every 70 minutes by overflowing the inlet with dry UHP $N_2$. Calibration factors   were determined by  the in-field continuous addition of a C-13 isotopically labelled formic acid standard to the tip of the inlet. Laboratory calibrations of the Ox-CIMS to formic acid and $O_3$ as a function of specific humidity were determined immediately pre- and post-campaign and were used to calculate a humidity dependent sensitivity of $O_3$ relative to formic acid. That relative sensitivity was then used to determine the in-field sensitivity to $O_3$ by scaling field sensitivities of formic acid from the continuous additions. Full details of this deployment and calibration methods are described in Vermeuel et al., (2019)."

*L356-358: How do these temperatures impact instrument performance? Is it species dependent?*

The primary motivation for these temperatures was to ensure that the inlet and instrument front end temperatures were always higher than ambient temperatures to prevent any condensation of water vapor. $O_3$ is typically considered a "non-sticky" molecule and is likely not significantly impacted by the selected temperature as long as it is ensured that there is no condensation of water.

*L429: Does 84*

It appears the Reviewers comment here was lost somewhere in the uploading process but we will provide additional detail here to hopefully answer their question. The 84% rejected flux periods refers to all ambient sampling periods, including those where winds were not from the ocean. The wind direction filter removed approximately 30% of the campaign data, followed by an additional 57% reduction due to the $u_*$ filter and further cuts due to the stationarity and outlier filters.

Additionally, please note that all values in this section have been revised as described in Comment 1 to all reviewers.

L431: How does 'despiking' impact the results?
The despiking algorithm as applied is intended to remove short large spikes in the data, primarily driven by electronics issues. If these short electronic spikes were left in the data they could bias the observed flux value and increase the LOD, as they would drive a strong short covariance signal. This despiking correction and our specific implementation are standard in EC data processing. Such spikes were negligible in our dataset and the correction was applied for completeness.

L520-521: For the non-expert reader it may be useful to clarify whether this bias is specific to O3 measurement with Ox-CIMS, or to CIMS measurements of trace gases in general.
We agree that this is a useful point to clarify. This correction would impact all CIMS instruments as they also measure mixing ratios relative to moist air with variable density. Due to the high mixing ratios of $O_3$ and the small deposition magnitude to water, measurements of $O_3$ air-sea exchange are particularly sensitive to this potential bias. Still, this factor should be considered for all flux sensors that do not directly measure mole fractions or mixing ratios relative to dry air.

The manuscript has been revised to state: Line 520: "The Ox-CIMS measures $O_3$ as the apparent mixing ratio relative to moist air, as is true of all CIMS based measurements, which means fluctuations in the density of air due to changes in temperature, pressure, and humidity could introduce a bias in the EC flux measurement (Webb et al., 1980)."

L766-767: incomplete citation to Vermeuel 2019. doi?
Thank you, this citation has been corrected.

**Technical Corrections:**
L194: "was seen to have" to "had"
L223: "analytes analyzed" to "analytes detected/measured"
L305: Repeated section title? (Same as 2.9)
L527: Repeated phrase "which is removed by active heating of the inlet"
L528: semicolon use
L590: validation to validate

We thank the reviewer for their careful reading! All of the above technical corrections have been made following the suggestions of the reviewer.

The title of Section 2.10 was indeed accidental copied from the title of Section 2.9. The title of Section 2.10 has been corrected to: "Short- and long- term precision"

Responses to Reviewer 3:

*General comments:*
*A novel method of measuring O3 and NO2 based on chemical ionization time-of-flight mass spectrometry with oxygen anion (O2-) as the reagent ion (Ox-CIMS) is developed. This new method is able to measure O3 and NO2 at fast time response and low mixing ratios, which is applicable to eddy covariance flux measurements. The authors conducted thorough characterization of the sensitivity, ion chemistry, inlet, calibration in the laboratory. They also used the instrument for the measurement of O3 vertical fluxes over the coastal ocean, via eddy covariance. Their measured flux is in good agreement with prior studies of O3 ocean-atmosphere exchange. Potentially, fluxes for multiple species can be obtained with one measurement with the Ox-CIMS. During the same campaign, they also used a 2B ozone monitor to measure ozone, which agreed well with the Ox-CIMS measurement. The paper is well written, and I suggest publishing this work after addressing the following specific comments.*

We thank the reviewer for their thoughtful and supportive review. Responses to specific reviewer comments are below.

Specific comments:
1.) *Around line 138 to 153 on the discussion of CO3- ion formation, do other chemicals also form CO3-? It was mentioned early on line 119 that SO2 also forms CO3-? How to rule out that CO3- detected are not from other chemicals? Similarly, on line 215, would CO3- come from other species, rather than O3+CO2+O2-chemistry?*

We believe the discussion of use of CO3- reagent ions for detection of SO2 was confusingly worded. In that work, SO2 does not generate CO3-, rather CO3- reagents are used to ionize SO2 forming SO3-. We highlighted this prior work as CO3- reagent ions were made by first generating O2- ions and reacting them with intentionally added O3 in the presence of CO2. In our work the same mechanism is used, but rather than intentionally adding excess O3 to form CO3-, we detect ambient O3 in the form of the CO3- product.

We believe it is unlikely that other species significantly compete with the O3+CO2+O2-chemistry. For a species to favorably transfer an O- to CO2 it must be a strong gas phase oxidizing agent and form a more stable product following O- transfer. It is apparent what species might follow the above requirements. Those species if present are also likely highly reactive oxidized molecules which would be at low mixing ratios relative to O3.

The manuscript has been revised to state: starting at line 118 " Oxygen anion chemistry has also been used for the detection of $SO_2$ *via* a multi-step ionization process where $CO_3^-$ reagent ions are first generated by the reaction of $O_2^-$ with added excess $O_3$ in the presence of $CO_2$. The $CO_3^-$ reagent ion  then ligand switches with $SO_2$ to form $SO_3^-$ which then quickly reacts with ambient $O_2$ to form the primary detected $SO_5^-$ product (Porter et al., 2018; Thornton et al., 2002a)."

2.) *Line 154: Are there other interfering species that will end up as NO2- in the CIMS? Do HNO3, HONO, PAN or Organ-NO2 form NO2- with the ion chemistry? For example, on line 282, the authors mention that "A possible source of this background is from degradation of other species such as nitric acid or alkyl nitrates on the inlet walls." Did the authors do any test for interfering species?*

We agree that this is an important consideration that should be made clear to the reader. We have not yet extensively tested for other potential interfering species at the NO2- product. Laboratory and field calibrations of HNO3 do not show contribution to the NO2- product, but a small fraction is seen as NO3-. The conservation of total odd-oxygen during an O3 titration event discussed in Section 4 gives some qualitative indication that interfering species are likely small, but more direct evidence is necessary. Further validation of the NO2 detection product is certainly warranted to characterize any interferences but we believe that to be beyond the scope of this work. The manuscript focuses primarily on the O3 measurement and we tried to be upfront that the NO2 measurement currently has less validation than the O3 measurement. We make further reference to this in section 5 (Conclusions and outlook) stating "Further optimization and characterization of the Ox-CIMS is ongoing, including efforts to validate the specificity of the $NO_2$ detection…"

The manuscript has been revised to state: Line 282: "Additional calibration will be necessary to ensure that observed $NO_2$ signal is not a secondary product of other species and we cannot currently quantify their potential interference on measured $NO_2$."

3.) *Line 189, can the authors specify what the normalized counts are? Is it normalized to the reagent ion counts?*
Based on suggestions from Reviewers 2 and 3, discussion of the normalization of the normalization process has been clarified. Generally, we use it in the same was as is common in chemical ionization mass spectrometry applications, where all signals are scaled relative to a fixed total reagent ion signal of $1 \times 10^6$ cps.

The manuscript has been revised to state: Line 188: "Sensitivity values can be normalized by scaling all signals to a fixed total reagent ion signal of $1 \times 10^6$ cps to isolate the sensitivity component controlled by reagent ion chemistry, separate from  changes in instrument performance due to decay in the ion source or other factors. The total reagent ion signal is taken as the sum of the $O_2^-$ and $O_2(H_2O)^-$ signals."

4.) *Section 2.8: The authors mentioned that background measurement influences the detection limit. Do they have any recommendation in improving the detection limit?*

Because the instrument O3 background is driven by chemistry in the reagent-ion source it is not clear what best approach is for reducing this background. We speculate that use of an alternative ionization source (*i.e.* soft x-rays or a corona discharge) may reduce this background issue but that would require laboratory work beyond the scope of this work. Alternatively a mass selective filter at the interface of the ion source and the IMR could be

used to remove the larger $O_3^-$ ions (-m/$Q$ 48) and preserve the $O_2^-$ reagent ions (-m/$Q$ 32) which would also be a substantial undertaking.

5.) *Line 572: It might be easier for readers to include the equation in the paper and cite Bariteau et al., so readers won't need to download Bariteau et al.*

The equations from Bariteau. 2010 have been added to the text as suggested. In the course of revising the text, we realized a citation of Lenschow and Kristensen (1985) was also warranted, which has also been added.

The manuscript has been revised in this section to list the equations and define all variables starting at Line 572.

References:

Lenschow, D. H. and Kristensen, L.: Uncorrelated Noise in Turbulence Measurements, J. Atmos. Ocean. Technol., 2, 68–81, doi:https://doi.org/10.1175/1520 0426(1985)002<0068:UNITM>2.0.CO;2, 1985.